# Distributed representations of behaviour-derived object dimensions in the human visual system

Oliver Contier [1,2] ✉, Chris I. Baker [3] & Martin N. Hebart [1,4]

Object vision is commonly thought to involve a hierarchy of brain regions processing increasingly complex image features, with high-level visual cortex supporting object recognition and categorization. However, object vision supports diverse behavioural goals, suggesting basic limitations of this category-centric framework. To address these limitations, we mapped a series of dimensions derived from a large-scale analysis of human similarity judgements directly onto the brain. Our results reveal broadly distributed representations of behaviourally relevant information, demonstrating selectivity to a wide variety of novel dimensions while capturing known selectivities for visual features and categories. Behaviour-derived dimensions were superior to categories at predicting brain responses, yielding mixed selectivity in much of visual cortex and sparse selectivity in category-selective clusters. This framework reconciles seemingly disparate findings regarding regional specialization, explaining category selectivity as a special case of sparse response profiles among representational dimensions, suggesting a more expansive view on visual processing in the human brain.

A central goal of visual neuroscience is to understand how the brain encodes and represents rich information about objects, allowing us to make sense of our visual world and act on it in meaningful ways. A widely studied and influential account posits that one central function of the visual system is to recognize objects by organizing them into distinct categories[1–4]. According to this view, early visual cortex serves to analyse incoming visual information by representing basic visual features[5], which are then combined into more and more complex feature combinations, until higher-level visual regions in the occipitotemporal cortex and beyond support the recognition of object identity and category[3]. In line with this view, a number of category-selective clusters have been identified in occipitotemporal cortex that respond selectively to specific object classes such as faces, scenes, body parts, tools or text[6–11]. The functional importance of these regions is underscored by studies demonstrating that object category and identity as

well as performance in some behavioural tasks can be read out from activity in occipitotemporal cortex[12–17] and that lesions to these regions can lead to selective deficits in object recognition abilities[18–22].

Despite the importance of object categorization and identification as crucial goals of object vision, it has been argued that these functions alone are insufficient for capturing how our visual system allows us to make sense of the objects around us[23]. A more comprehensive understanding of object vision should account for the rich meaning and behavioural relevance associated with individual objects beyond discrete labels. This requires incorporating the many visual and semantic properties of objects that underlie our ability to make sense of our visual environment, perform adaptive behaviours and communicate about our visual world[23–27]. Indeed, others have proposed that visual cortex is organized on the basis of continuous dimensions reflecting more general object properties, such as animacy[28–31], real-world

[1]Vision and Computational Cognition Group, Max Planck Institute for Human Cognitive and Brain Sciences, Leipzig, Germany. [2]Max Planck School of Cognition, Leipzig, Germany. [3]Laboratory of Brain and Cognition, National Institute of Mental Health, National Institutes of Health, Bethesda, MD, USA. [4]Department of Medicine, Justus Liebig University Giessen, Giessen, Germany. ✉e-mail: contier@cbs.mpg.de

size[29,32], aspect ratio[31,33] or semantics[34]. These and other continuous dimensions reflect behaviourally relevant information that offers a more fine-grained account of object representations than discrete categorization and recognition alone. This dimensional view suggests a framework in which visual cortex is organized on the basis of topographic tuning to specific dimensions that extends beyond category-selective clusters. Under this framework, category-selective clusters may emerge from a more general organizing principle[34–38], reflecting cortical locations where these tuning maps encode feature combinations tied to specific object categories[34,38,39]. Yet, while previously proposed dimensions have been shown to partially reflect activity patterns in category-selective clusters[40–45], they cannot account fully for the response profile and are largely inferior to category selectivity in explaining the functional selectivity of human visual cortex for objects[46,47].

To move beyond the characterization of individual behavioural goals underlying both the discrete category-centric and the continuous dimensional views and to comprehensively map a broad spectrum of behaviourally relevant representations, one powerful approach is to link object responses in visual cortex to judgements about the perceived similarity between objects[48–51]. Indeed, perceived similarity serves as a common proxy of mental object representations underlying various behavioural goals, as the similarity relation between objects conveys much of the object knowledge and behavioural relevance across diverse perceptual and conceptual criteria[52–56]. Perceived similarity is therefore ideally suited for revealing behaviourally relevant representational dimensions and how these dimensions are reflected in cortical patterns of brain activity.

To uncover the nature of behaviourally relevant selectivity underlying similarity judgements in human visual cortex, in the present study we paired functional MRI (fMRI) responses to thousands of object images[57] with core representational dimensions derived from a dataset of millions of human similarity judgements. In contrast to much previous research that has focused on a small number of hypothesis-driven dimensions or that used small, selective image sets[29,48–51,58–60], we carried out a comprehensive characterization of cortical selectivity in response to 66 representational dimensions identified in a data-driven fashion for 1,854 objects[52,61].

Moving beyond the view that mental object representations derived from similarity judgements are primarily mirrored in high-level visual cortex[48–50,57], we demonstrate that representations underlying core object dimensions are reflected throughout the entire visual cortex. Our results reveal that cortical tuning to these dimensions captures the functional topography of visual cortex and mirrors stimulus selectivity throughout the visual hierarchy. In this multidimensional representation, category selectivity stands out as a special case of sparse selectivity to a set of core representational object dimensions, while other parts of visual cortex reflect a more mixed selectivity. A direct model comparison revealed that continuous object dimensions provide a better model of brain responses than categories across the visual system, suggesting that dimension-related tuning maps offer more explanatory power than a category-centric framework. Together, our findings reveal the importance of behaviour-derived object dimensions for understanding the functional organization of the visual system and offer a broader, comprehensive view of object representations that bridges the gap between regional specialization and domain-general topography.

## Results

We first aimed at mapping core representational object dimensions to patterns of brain activity associated with visually perceived objects. To model the neural representation of objects while accounting for their large visual and semantic variability[62,63], we used the THINGS-data collection[57], which includes densely sampled fMRI data for thousands of naturalistic object images from 720 semantically

diverse objects, as well as 4.7 million behavioural similarity judgements of these objects (Fig. 1).

As core object dimensions, we used a recent similarity embedding of behaviour-derived object dimensions, which underlie the perceived similarity of 1,854 object concepts[52,57]. In this embedding, each object image is characterized by 66 dimensions derived from the human similarity judgements in an odd-one-out task. We chose this embedding for several reasons. First, it provides highly reproducible dimensions that together are sufficient for capturing single-trial object similarity judgements close to the noise ceiling. Second, the use of an odd-one-out task supports the identification of the minimal information required to distinguish between different objects and thus is sensitive not only to conceptual information, such as high-level category (for example, 'is an animal'), but also to key visual–perceptual distinctions (for example, 'is round'). The object dimensions thus capture behaviourally relevant information, in that they support the key factors underlying arbitrary categorization behaviour and therefore underlie our ability to make sense of our visual world, to generalize, to structure our environment and to communicate our knowledge. Indeed, the object dimensions capture external behaviour such as high-level categorization and typicality judgements, underscoring their potential explanatory value as a model of neural responses to objects[52]. Third, the object dimensions are easily interpretable, thus simplifying the interpretation of neural activity patterns in relation to individual dimensions.

The fMRI dataset covers 8,740 unique images from 720 categories presented to three participants (two female) over the course of 12 sessions[57]. Given that the behavioural similarity embedding was trained only on one image for each of the 1,854 THINGS categories, these dimensions may only partially capture the visual richness of the entire image set, which may affect the potential for predicting image-wise brain responses. To address this challenge, we fine-tuned the artificial neural network model CLIP-ViT[64] to directly predict object dimensions for the 8,740 images in our fMRI dataset. This model has previously been shown to provide a good correspondence to behavioural[65,66] and brain data[67,68], indicating its potential for providing accurate image-wise estimates of behaviour-derived object dimensions. Indeed, this prediction approach led to highly accurate cross-validated predictions of object similarity[69] and consistent improvements in blood-oxygen-level-dependent (BOLD) signal predictions for all 66 dimensions (Supplementary Fig. 1).

### Core object dimensions are reflected in widespread fMRI activity patterns throughout the human visual system

To test how these dimensions were expressed in voxel-wise brain responses, we fit an fMRI encoding model that predicts spatially resolved brain responses on the basis of a weighted sum of these object dimensions. This allowed us to map out the contribution of the dimensions to the measured signal and thus link interpretable behaviour-derived dimensions to patterns of brain activity.

Across all 66 object dimensions, our results revealed a widely distributed cortical representation of these dimensions that spans much of visual cortex and beyond (Fig. 2). The spatial extent of these effects was highly similar across all three participants, underscoring the generality of these findings. We also tested the replicability of these results on an independent fMRI dataset[70], revealing a similarly extensive representation of the object dimensions (Supplementary Fig. 2). Please note that, in the following, we use the terms 'widespread' and 'distributed' interchangeably and do not refer to a distributed representational coding scheme or the presence of continuous gradients but rather to responses that are not locally confined.

Prediction accuracies not only peaked in lateral occipital and posterior ventral temporal regions but also reached significant values in early visual, dorsal visual and frontal regions (Supplementary Fig. 3). In contrast to previous work based on representational similarity analysis that found information about perceived similarity to be confined

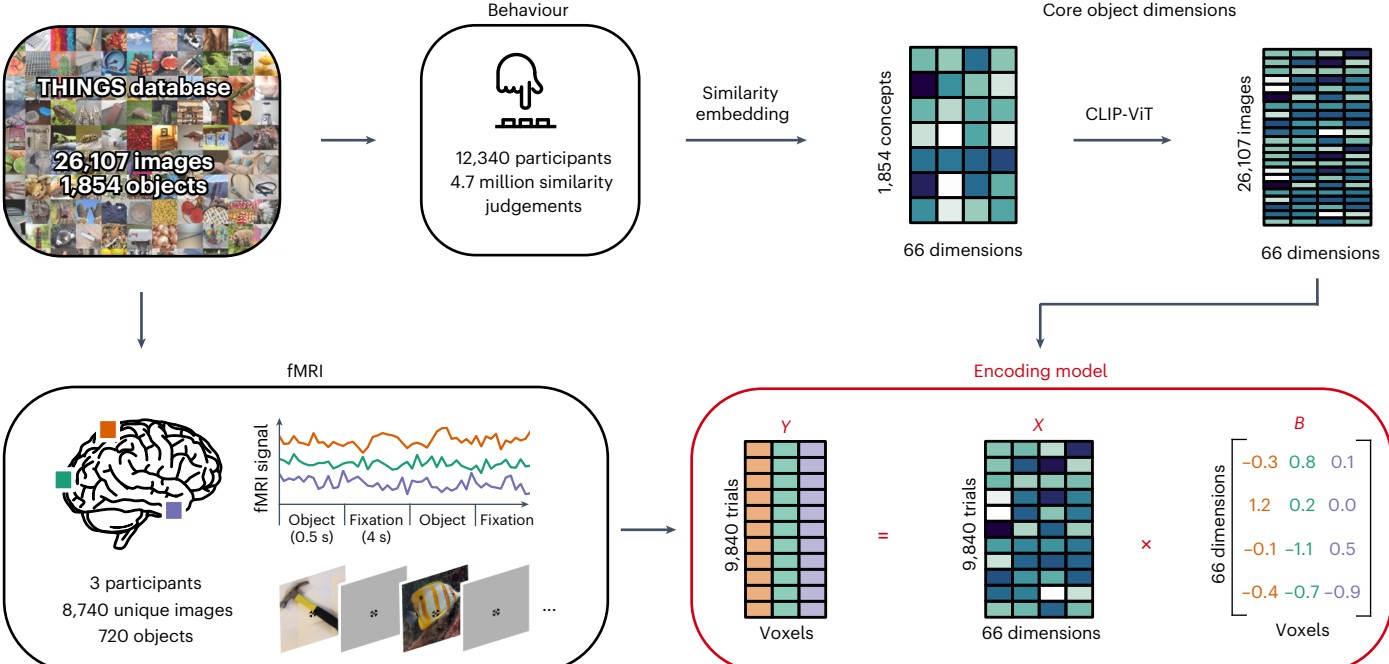

**Fig. 1 | An fMRI encoding model of object dimensions underlying human similarity judgements.** We linked core representational dimensions capturing the behavioural relevance of objects to spatially resolved neural responses to thousands of object images. For this, we used the THINGS-data collection[57], which includes fMRI and behavioural responses to objects from the THINGS object concept and image database[82]. The behavioural data were used to train a computational model of core object dimensions underlying human similarity judgements on different object concepts. We extended this embedding to the level of individual object images on the basis of the computer vision model CLIP-ViT[64]. The fMRI data comprise three participants who each saw 8,740 unique object images. We used an encoding model of the object dimension embedding to predict fMRI responses to each image in each voxel. The estimated encoding model weights reflect the tuning of each voxel to each object dimension. $X$, $B$ and $Y$ denote the design matrix, regression weights and outcome of the encoding model, respectively.

primarily to higher-level visual cortex[49–51,57], our dimension-based approach revealed that behaviourally relevant information about objects is much more distributed throughout the visual processing hierarchy, including the earliest cortical processing stages.

**Behaviour-derived object dimensions reflect the functional topography of the human visual system**

Having identified where information about perceived similarity is encoded, we next explored the spatial layout of each individual dimension underlying this representation. By using a voxel-encoding model of interpretable object dimensions, it is possible to inspect the cortical distribution of the weights of each regressor separately and interpret them in a meaningful fashion. This has two benefits. First, it allows us to probe to what degree behaviour-derived dimensions alone can capture the known topography of visual cortex. Second, it allows us to identify novel topographic patterns across visual cortex. This provides important insights into how the topography of visual cortex reflects object information relevant to behaviour and how functionally specialized regions are situated in this cortical landscape.

Visualizing the voxel-wise regression weights for each object dimension on the cortical surface (Fig. 3) revealed a clear correspondence between numerous dimensions and characteristic, known topographic patterns of the visual system. For example, the 'animal-related' dimension mirrors the well-established spoke-like tuning gradient for animate versus inanimate objects[29], while dimensions such as 'head-related' and 'body-part-related' differentiate the regional selectivity for faces and body parts in the fusiform face area (FFA), occipital face area (OFA) and extrastriate body area (EBA)[6,7,71]. Likewise, the implicit inclusion of natural scenes as object backgrounds revealed scene-content-related dimensions (for example, 'house/furnishing-related', 'transportation/movement-related' and 'outdoors'), which were found to be associated with scene-selective brain regions such as parahippocampal place area (PPA), medial place area (MPA) and occipital place area (OPA)[8,72–76]. Our approach also independently identified a 'food-related' dimension in areas adjacent to the fusiform gyrus, in line with recently reported clusters responding selectively to food stimuli[77–79]. A dimension related to tools ('tool-related/handheld/elongated') also matched expected activation patterns in middle temporal gyrus[11,80,81]. Furthermore, dimensions related to low-to mid-level visual features (for example, 'grid/grating-related' and 'repetitive/spiky') reflected responses primarily in early visual cortex.

Beyond these established topographies, the results also revealed numerous additional topographic patterns. For example, one dimension reflected small, non-mammalian animals ('bug-related/non-mammalian/disgusting') that was clearly distinct from the 'animal-related' dimension by lacking responses in face and body selective regions. Another dimension reflected a widely distributed pattern in response to thin, flat objects ('thin/flat/wrapping'). Our approach thus allowed for the identification of candidate functional selectivities in visual cortex that might have gone undetected with more traditional approaches based on proposed categories or features[47,77]. Importantly, the functional topographies of most object dimensions were also found to be highly consistent across the three participants in this dataset (Supplementary Fig. 4) and largely similar to participants in an independent, external dataset (Supplementary Fig. 2), suggesting that these topographies may reflect general organizing principles rather than idiosyncratic effects (Supplementary Fig. 4 and Extended Data Figs. 1–6).

Together, our results uncover cortical maps of object dimensions underlying the perceived similarity between objects. These maps span extensive portions of the visual cortex, capturing topographic characteristics such as tuning gradients of object animacy, lower-level

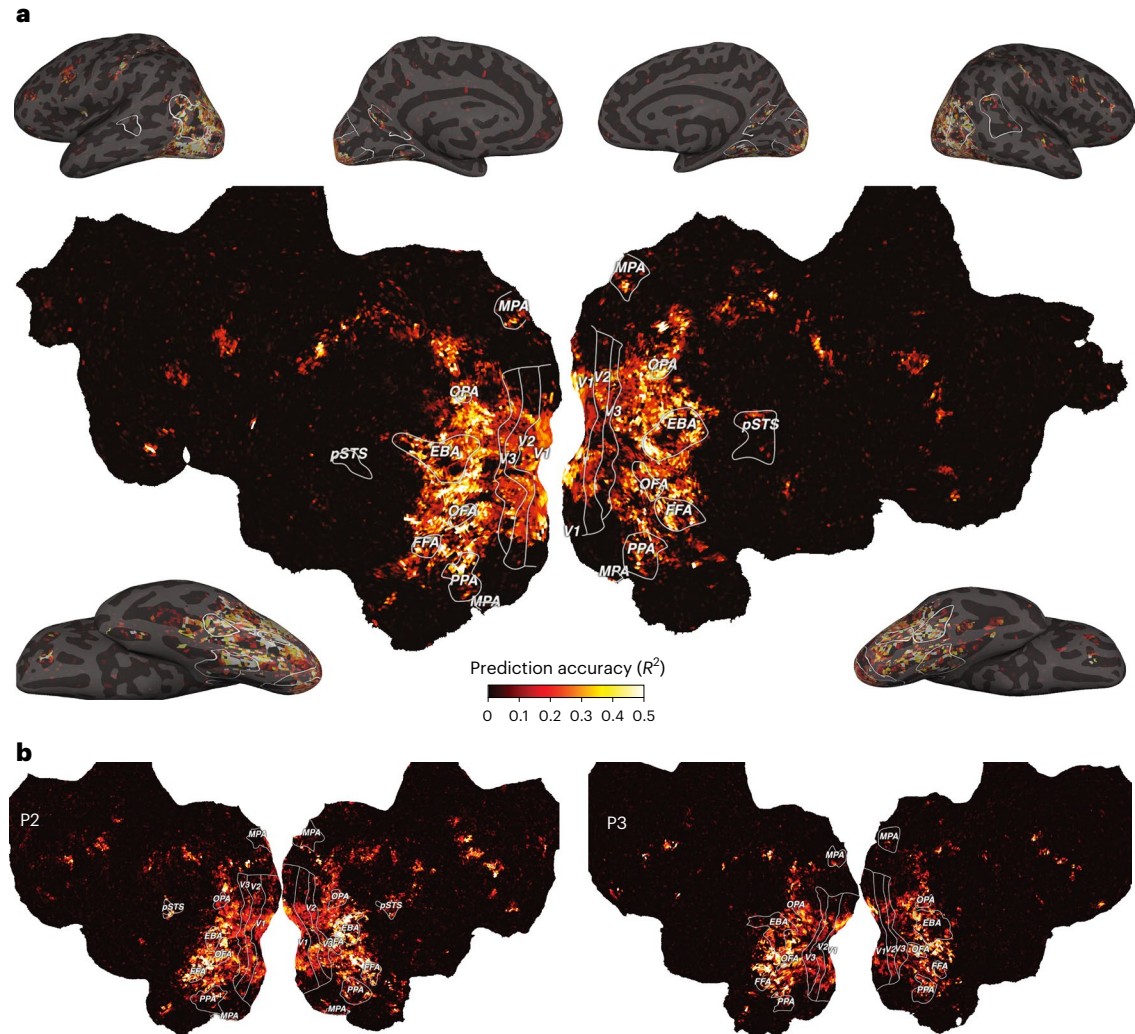

**Fig. 2 | Prediction accuracy of the fMRI voxel-wise encoding model based on 66 core object dimensions. a**, Prediction accuracy for one example participant (P1) visualized on a cortical flat map (centre) and inflated views of the cortical surface (corners). **b**, Results for the other two participants visualized on cortical flat maps. The colours indicate the proportion of explained variance (noise-ceiling-corrected $R^2$) of held-out data in a 12-fold between-session cross-validation. The white outlines indicate regions of interest defined in separate localizer experiments: FFA, OFA, posterior superior temporal sulcus (pSTS), EBA, PPA, OPA, MPA and V1–V3.

visual feature tuning in early visual cortex and category-selective, higher-level regions while uncovering new candidate selectivities. These findings thus support an organizing principle where multiple, superimposing cortical tuning maps for core object properties collectively represent behaviourally relevant information about objects.

### Cortical tuning to behaviour-derived object dimensions explains regional functional selectivity

Having delineated the multidimensional topographic maps across visual cortex, we next homed in on individual brain regions to determine their functional selectivity as defined by their response tuning across these behaviour-derived dimensions. To this end, we developed a high-throughput method to identify object images representative for specific brain regions. Specifically, we first determined a functional tuning profile across dimensions for each region of interest based on the region's mean encoding model weights. Next, we identified images whose behavioural dimension profile best matched the functional tuning profile of the brain region. To this end, we used all 26,107 object images in the THINGS database[82], most of which were unseen by participants, and assessed the cosine similarity between the dimension profiles of brain regions and images. This enabled us to rank over 26,000 images on the basis of their similarity to a given brain region's functional tuning profile.

Despite having been fitted solely on the 66-dimensional similarity embedding, our approach successfully identified diverse functional selectivities of visual brain regions (Fig. 4). For instance, the most representative images for early visual regions (primary to tertiary visual cortex, V1–V3) contained fine-scale, colourful and repeating visual features, consistent with known representations of oriented edges and colour in these areas[83,84]. These patterns appeared more fine-grained in earlier (V1 or V2) than in later retinotopic regions (human V4, hV4), potentially reflecting increased receptive field size along the retinotopic hierarchy[85–87]. A similar finding is reflected in dimension selectivity profiles (Fig. 4), revealing higher colour selectivity in hV4 than in early retinotopic regions V1–V3 while yielding reductions in the 'repetitive/spiky' dimension. Notably, tuning profiles in category-selective regions aligned with images of expected object categories: faces in face-selective regions (FFA and OFA), body parts in body-part-selective regions (EBA) and scenes in scene-selective regions (PPA, OPA and MPA). Closer inspection of the tuning profiles revealed differences between regions that respond to the same basic object category, such as a stronger response to the 'body-part-related'

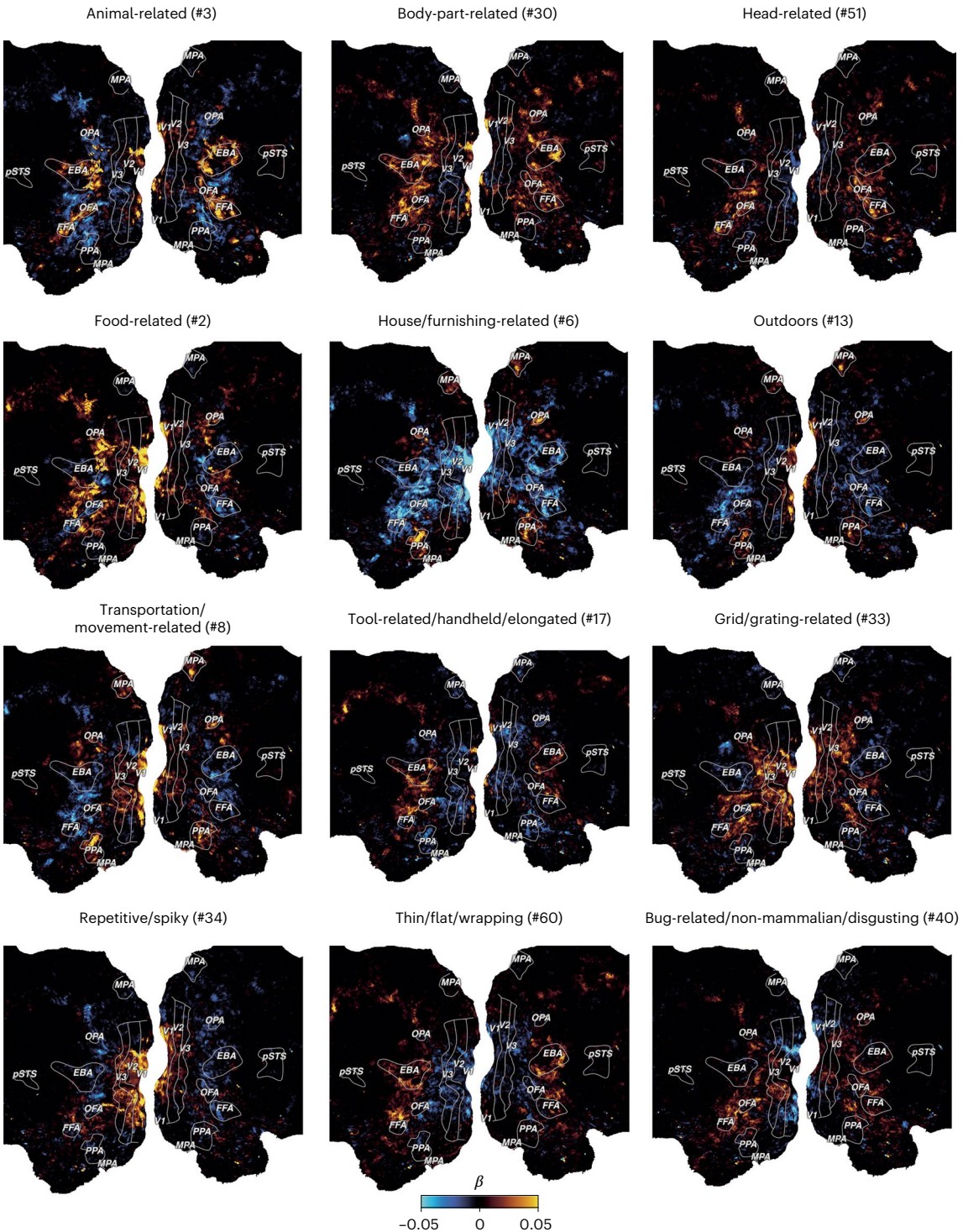

**Fig. 3 | Functional tuning maps to individual object dimensions.** Example maps for 12 of the 66 dimensions for participant P1. Each panel shows the encoding model weights for one object dimension projected onto the flattened cortical surface. The numbers in the panel labels show the dimension number in the embedding.

dimension in OPA but not in other place-selective regions. Also, selectivity to faces (FFA and OFA) versus body parts (EBA) appeared to be driven by the response magnitude to the 'head-related' dimension, while tuning to the remaining dimensions was highly similar across these regions. Together, these findings demonstrate that the 66 object dimensions derived from behaviour capture the selectivity across the visual processing hierarchy, highlighting the explanatory power of the dimensional framework for characterizing the functional architecture of the visual system.

## Category-selective brain regions are sparsely tuned to behaviour-derived object dimensions

Given that dimensional tuning profiles effectively captured the selectivity of diverse visual regions, we asked what factors distinguish category-selective visual brain regions from non-category-selective regions in this dimensional framework. We reasoned that category selectivity reflects a sparsely tuned representation, where activity in category-selective regions is driven by only a few dimensions, while non-category-selective regions reflect a more mixed selectivity, with

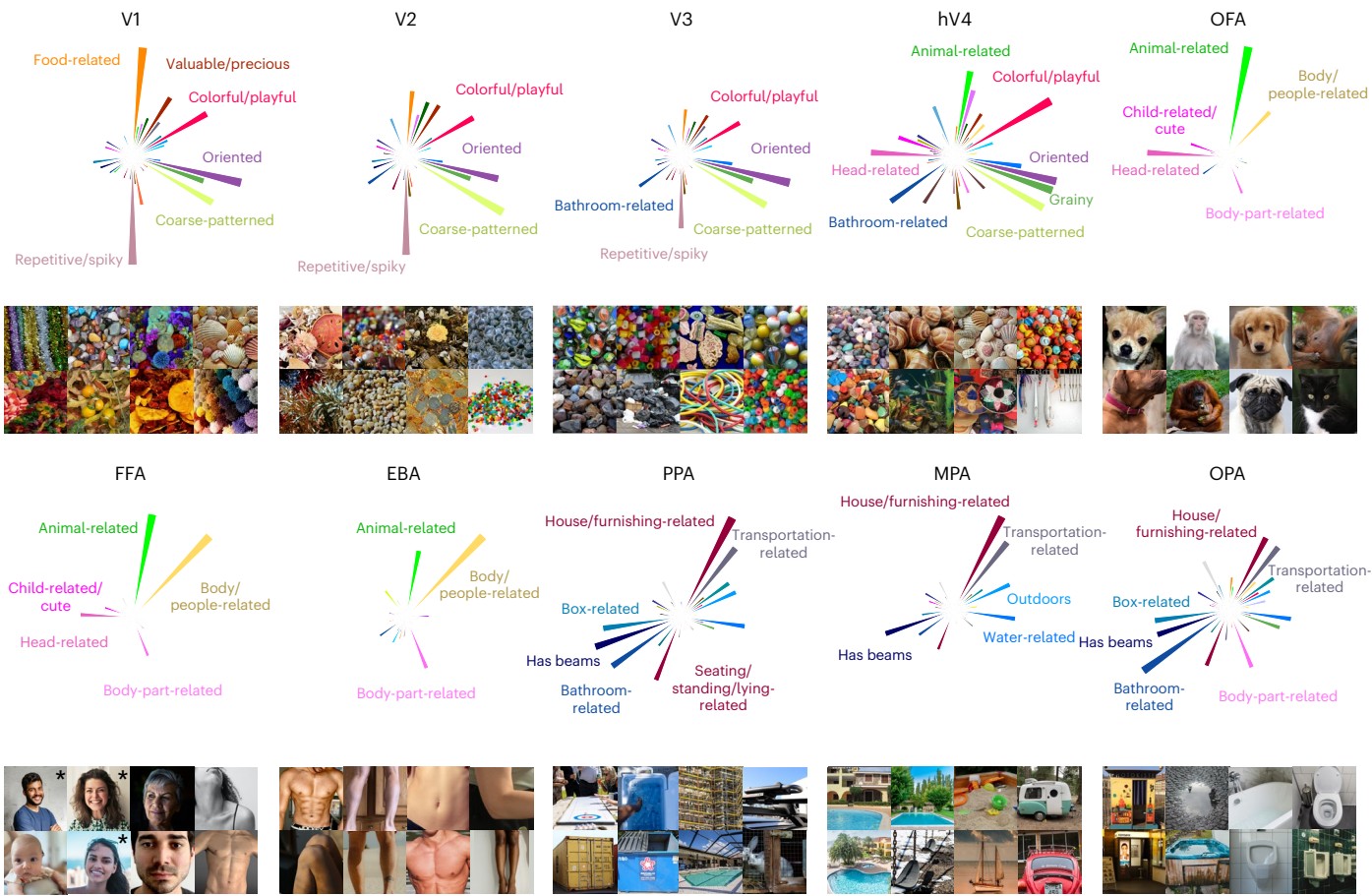

**Fig. 4 | Regional tuning profiles across 66 object dimensions and representative images for selectivity of each region of interest in visual cortex.** The rose plots indicate the magnitude of tuning for each object dimension in a given visual brain region. The image panels show eight images with the most similar model representation to the regional tuning profile. For copyright reasons, all original images have been replaced with visually similar images, and images of minors for which no permission could be obtained have been replaced with images of adults (marked with asterisks). The original images are available upon request. Photos from Pixabay.com and Pexels.com.

activity related to a larger number of dimensions. In this way, functionally specialized, category-selective regions might stand out as an extreme case of multidimensional tuning. As a consequence, this would also make it easier to identify category-selective regions due to their sparser selectivity.

To quantify this, we estimated a measure of sparseness over the encoding model weights in each voxel. Large sparseness indicates regions that are selective to very few dimensions, while lower sparseness indicates a dense representation in regions that respond broadly to diverse dimensions. Our results (Fig. 5a) indeed revealed sparser tuning in category-selective regions than in other parts of the visual system. This effect was most pronounced in face- and body-part-selective regions (FFA, OFA and EBA), with the sparsest tuning across all participants. The face-selective posterior superior temporal sulcus exhibited particularly sparse representation in Participants 1 and 2, while this region was not present in Participant 3 and, as expected, yielded no increase in sparseness. Scene-selective regions (PPA, MPA and OPA) also exhibited sparseness, though the effects were more variable across participants, which could arise from the representational dimensions being derived from objects within scenes, as opposed to isolated scene images without a focus on individual objects. Conversely, non-category-selective regions, such as early visual cortices, clearly exhibited dense representations. These findings suggest that category-selective regions, while responsive to multiple dimensions, may primarily respond to a small subset of behaviourally relevant dimensions. Thus, in a multidimensional representational framework,

category selectivity may reflect a special case of sparse tuning within a broader set of distributed dimension tuning maps.

Beyond the increased sparseness in functionally selective clusters, which had been defined in an independent localizer experiment[57], we explored to what degree we could use sparseness maps for revealing additional, potentially novel functionally selective regions. To this end, we identified two clusters with consistently high sparseness values across participants (Fig. 5b). One cluster was located in the right hemisphere anterior to anatomically defined area FG4 (ref. 88) and between the functionally defined FFA and anterior temporal face patch[89], with no preferential response to human faces in two of three participants in a separate functional localizer. The other cluster was located in orbitofrontal cortex, coinciding with anatomically defined area Fo3 between the olfactory and medial orbital sulci[90]. Having identified these clusters, we extracted regional tuning profiles and determined the most representative object images for each cluster. Inspection of the tuning profiles in these sparsely tuned regions revealed that their responses were best captured by images of animal faces for the region anterior to FFA and sweet food for orbitofrontal cortex (Fig. 5c). While the results in orbitofrontal cortex are in line with the motivational importance of rewarding foods and food representations in frontal regions[78,91–94], the selective response to animal faces in the cluster anterior to FFA deserves further study. By identifying regional response selectivity in a data-driven fashion[95], the results show that sparse tuning can aid in localizing functionally selective brain regions, corroborating the link between representational dimensions and regional selectivity.

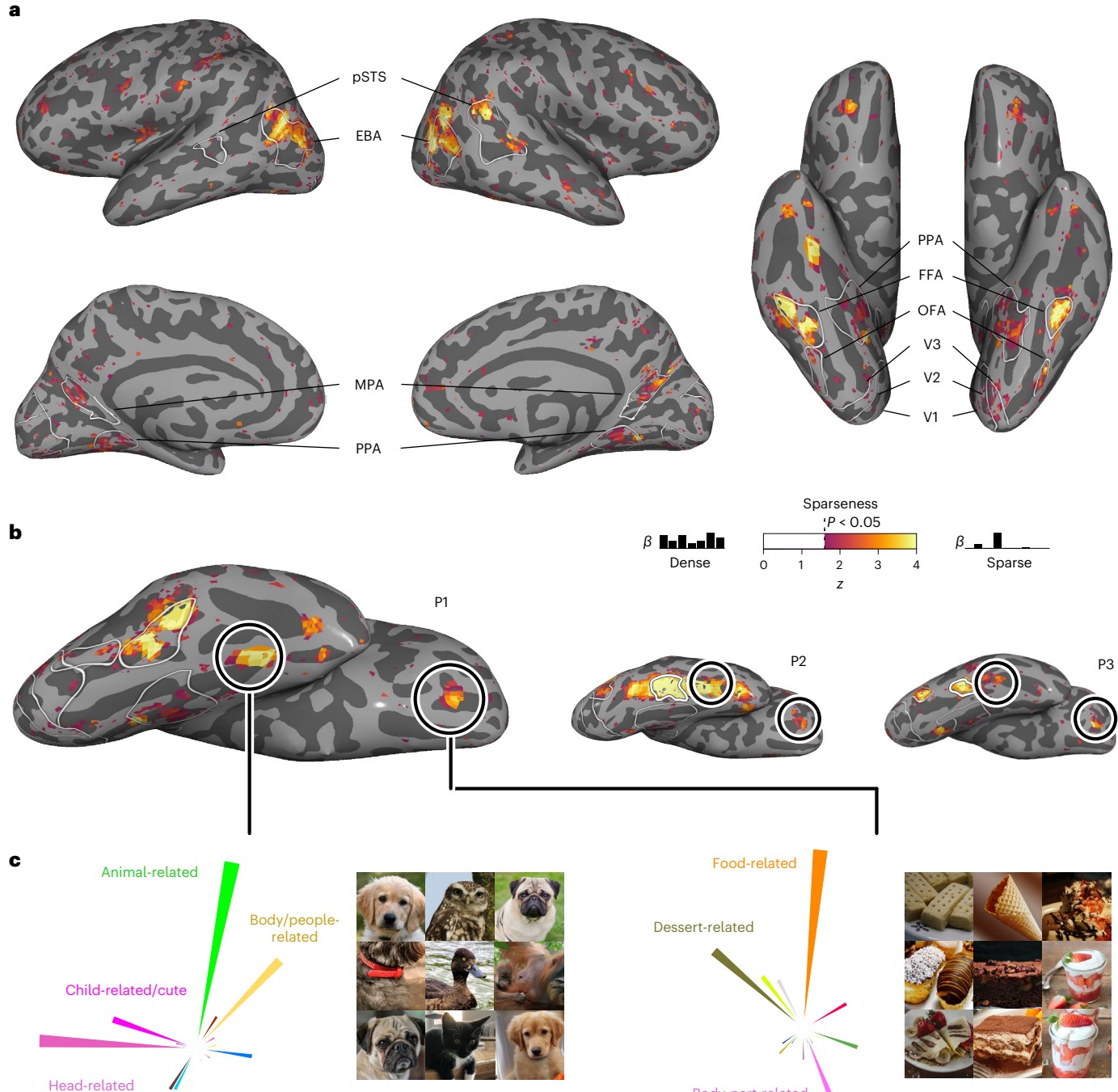

**Fig. 5 | Representational sparseness of behaviour-derived object dimensions in object-category-selective brain regions. a**, Inflated cortical surfaces for Participant 1 showing the sparseness over the encoding model weights in each voxel. The colours indicate z values of sparseness compared with a noise pool of voxels thresholded at $P < 0.05$ (one-sided, uncorrected). **b**, Ventral view of the right hemisphere for all three participants. The round outlines illustrate the locations of two explorative, sparsely tuned regions of interest: one in the fusiform gyrus and one in orbitofrontal cortex. **c**, Functional selectivity of these explorative regions of interest demonstrated by their multidimensional tuning profiles and most representative object images. For copyright reasons, all original images have been replaced with visually similar images. The original images are available upon request. Photos from Pixabay.com and Pexels.com.

## Object dimensions offer a better account of visual cortex responses than categories

If representational dimensions offer a better account of the function of ventral visual cortex than categorization, this would predict that they have superior explanatory power for brain responses to visually perceived objects in these regions[47,96]. To compare these accounts formally, we compiled a multidimensional and a categorical model of object responses and compared the amount of shared and unique variance explained by these models (for an exploratory comparison with object shape, see Supplementary Fig. 6 and Supplementary Methods 2). We first constructed a category model by assigning all objects appearing in the presented images into 50 common high-level categories (for example, 'animal', 'bird', 'body part', 'clothing', 'food', 'fruit' and 'vehicle') available as part of the THINGS metadata[97]. To account for the known selectivity to faces and body parts, we additionally labelled images in which faces or body parts appeared and included them as two additional categories. Then, for each category, we determined the most diagnostic object dimension.

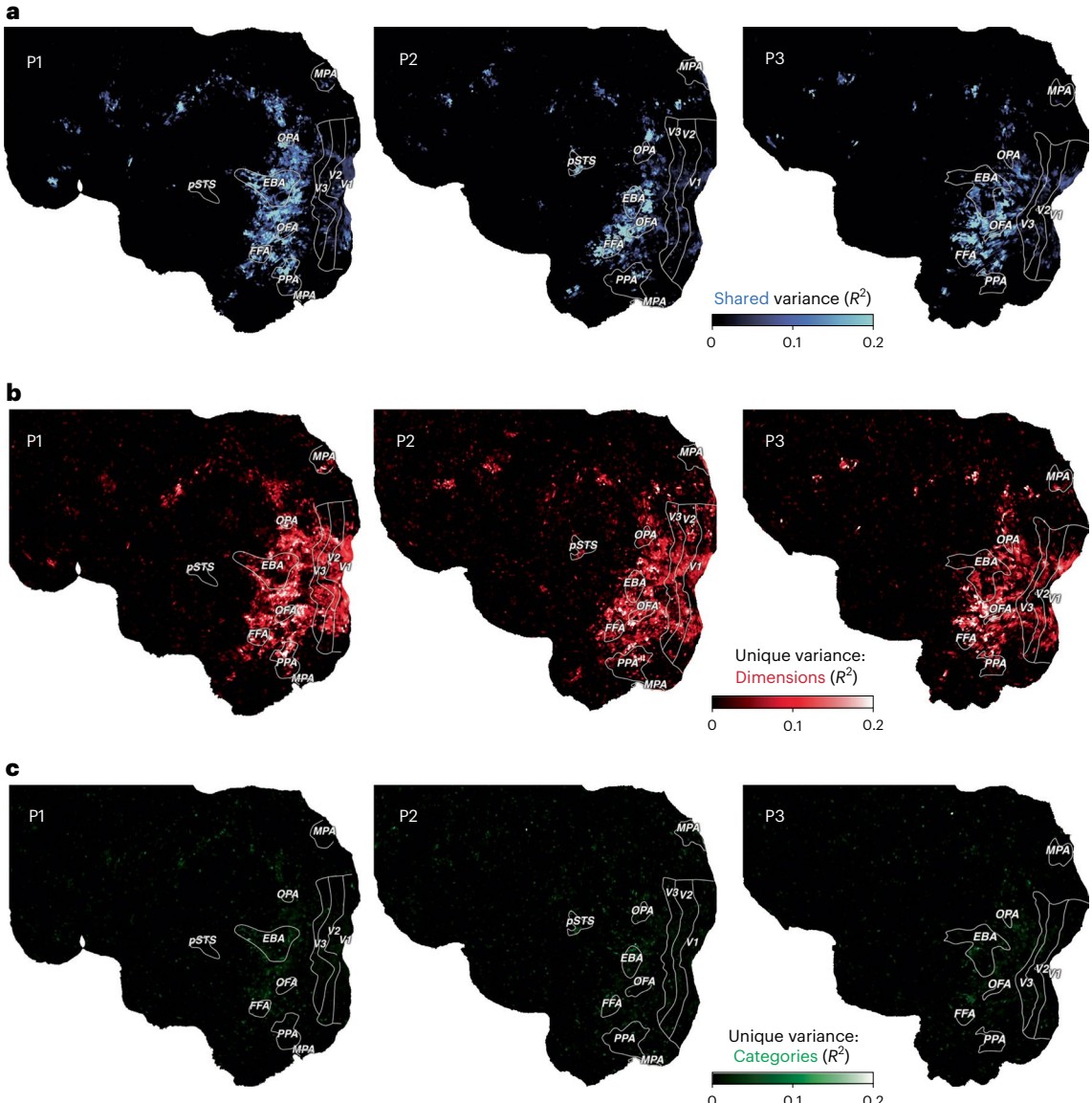

**Fig. 6 | Comparison of a continuous dimensional model and a categorical model of object responses. a**, Shared variance in single-trial fMRI responses explained by both models. **b**, Variance explained uniquely by a multidimensional model. **c**, Variance explained uniquely by a model of object categories. The flat maps show the left hemisphere of each participant. The colours indicate the proportion of explained variance (noise-ceiling-corrected $R^2$) from variance partitioning.

Since some dimensions mapped to multiple categories, this resulted in a model of 30 object dimensions. On the basis of the 52 categories and the 30 dimensions, we fit two encoding models to the fMRI single-trial responses and performed variance partitioning to disentangle the relative contributions of the object category and dimension models to the cross-validated prediction.

The results (Fig. 6) demonstrate that both object dimensions and categories shared a large degree of variance in explaining brain responses, especially in higher-level ventro-temporal and lateral occipital cortices (median, 19%; maximum, 74% shared explained variance) and to a lesser extent in early visual regions (median, 4%; maximum, 19% shared explained variance). This suggests that both models are well suited for predicting responses in the visual system. However, when we inspected the unique variance explained by either model, object dimensions explained a much larger amount of additional variance than object categories (Supplementary Fig. 5). This gain in explained variance was not only evident in higher-level regions (median, 10%; maximum, 35% unique explained variance), where both

models performed well, but extended across large parts of visual cortex, including early visual regions (median, 8%; maximum, 35% unique explained variance), suggesting that behaviour-derived dimensions captured information not accounted for by categories. Conversely, category membership added little unique explained variance throughout the visual system (median, 1%; maximum, 11%), reaching larger values in higher-level regions (median, 2%; maximum, 11% unique explained variance). Together, these results indicate that a multidimensional model offers an account with more explanatory value than a category model, supporting the idea that capturing behaviourally relevant responses in the visual system requires moving beyond categorization and suggesting object dimensions as a suitable model of encoding the behavioural relevance of objects.

## Discussion

Determining how the human brain represents object properties that inform our broad range of behaviours is crucial for understanding how we make sense of our visual world and act on it in meaningful ways.

Here we identified behaviour-derived brain representations by predicting fMRI responses to thousands of object images with 66 interpretable representational dimensions underlying millions of object similarity judgements. The results reveal that this behaviourally relevant information is mirrored in activity patterns throughout the entire visual processing hierarchy, emphasizing the importance of considering the entire system for identifying the behavioural relevance of visual responses. The diverse image selectivity of different visual brain regions emerged from the multidimensional tuning profiles in this distributed representation. This suggests that behaviour-derived dimensions offer a broadly applicable model for understanding the architecture of the visual system in which category-selective regions stand out as a special case of sparse tuning. A direct model comparison confirmed that such a multidimensional account has more explanatory value than a category-centric account.

Much work on the behavioural relevance of object responses in occipitotemporal cortex has focused primarily on a limited number of behavioural goals, such as recognition and high-level categorization[20–22,28,74,96]. According to this view, high-level visual regions contain representations that abstract from factors non-essential for recognition and categorization, such as position, colour or texture[3,98,99]. Our findings provide an alternative perspective on the nature of cortical object representations that may offer greater explanatory power than this traditional view. By considering a richer representation of objects supporting broader behavioural goals[23], object information is no longer restricted to the commonalities between objects based on how we label them. In this framework, even responses in early visual cortex to images from high-level categories such as food[77,78], which would traditionally be disregarded as lower-level confounds based on texture or colour, are relevant information supporting the processing of behaviourally relevant visual inputs. In this perspective, object vision solves the more general problem of providing a rich representation of the visual environment capable of informing a diverse array of behavioural domains[23].

While our results favour a distributed view of object representations, localized response selectivity for ecologically important object stimuli has been replicated consistently, underscoring the functional importance of specialized clusters. Regional specialization and distributed representations have traditionally been seen as opposing levels of description[37,38]. In contrast, our study advances a framework for unifying these perspectives by demonstrating that, compared with other visual regions, category-selective clusters exhibit sparse response tuning profiles. This framework treats regional specialization not as an isolated phenomenon but rather as a special case within a more general organizing principle. It thus provides a more general view of object representations that acknowledges the importance of regional specialization in the broader context of a multidimensional topography.

One limitation of our study is that we did not identify behaviour-derived dimensions specific to each individual participant tested in the MRI. Instead, dimensions were based on a separate population of participants. However, our findings were highly replicable across the three participants for most dimensions, suggesting that these dimensions reflect general organizing principles rather than idiosyncratic effects (Supplementary Fig. 4). Of note, some dimensions did not replicate well (for example, 'feminine (stereotypical)', 'hobby-related' or 'foot/walking-related'; Supplementary Fig. 4), which indicates that our fitting procedure does not yield replicable brain activity patterns for any arbitrary dimension. Future work may test the degree to which these results generalize to other dimensions identified through behaviour. Additionally, applying our approach to an external fMRI dataset (Supplementary Methods 1) revealed similarly distributed responses, with highly similar dimension tuning maps, suggesting that our findings generalize to independent participants (Supplementary Fig. 2). Future work could test the extent to which these results generalize to the broader population and how they vary between individuals. Furthermore, despite the broad diversity of objects used in the present study, our work excluded non-object images such as text[82]. While the effects of representational sparseness were less pronounced in scene-selective regions and largely absent in text-selective regions[10], our encoding model significantly predicted brain responses in scene-selective regions (Supplementary Fig. 3), indicating validity beyond isolated objects. Future research may extend these insights by exploring additional image classes. Moreover, our use of a pre-trained computational model[64] to obtain predicted dimension values might have underestimated the performance of the object embedding in predicting brain responses or may have selectively improved the fit of some dimensions more than that of others. Future studies could test whether using empirically measured dimension values for each image would lead to refined dimension maps. Finally, we reported results based on noise-ceiling-corrected $R^2$ values. While noise-ceiling normalization is common practice when interpreting encoding model results to make them more comparable, the degree to which the results would generalize if noise ceilings were much higher could probably only be addressed with much larger yet similarly broad datasets.

While the behaviour-derived dimensions used in this study were highly predictive of perceived similarity judgements and object categorization[52], there are many possible behaviours not captured by this approach. Here we used representational dimensions underlying similarity judgements to contrast with the category-centric approach. We chose similarity judgements as a common proxy for mental object representations, since they underlie various behavioural goals, including categorization and recognition[52–56]. Future work could test the extent to which other behaviours or computational approaches carry additional explanatory value[15,49,51,100,101]. This would also allow establishing the causal relevance of these activity patterns in behavioural readout[13,15,17,102].

Given the explanatory power of our dimensional framework, our results may be interpreted as hinting at an alternative explanation of traditional stimulus-driven feature selectivity through the lens of behavioural relevance[103], where the emergence of feature selectivity may exist because of the potential for efficient behavioural readout. Since the dimensions used in this study probably do not capture all behaviourally relevant selectivity, our approach does not allow testing this strong assumption. For example, a direct comparison of our embedding with the predictive performance of a Gabor wavelet pyramid model[104] or state-of-the-art deep neural network models[68] would neither support nor refute this idea. Future work could specifically target selectivity to individual visual features to determine the degree to which these representations are accessible to behavioural readout and thus may alternatively be explained in terms of behavioural relevance, rather than feature selectivity.

In conclusion, our work provides a multidimensional framework that aligns with the rich and diverse behavioural relevance of objects. This approach promises increased explanatory power relative to a category-centric framework and integrates regional specialization within a broader organizing principle, thus offering a promising perspective for understanding how we make sense of our visual world.

## Methods

### THINGS-data

We relied on the openly available THINGS-data collection to investigate the brain representation of everyday objects[57]. THINGS-data include 4.7 million human similarity judgements as well as neural responses measured with fMRI to thousands of naturalistic and broadly sampled object images. The collection also includes a representational embedding of core object dimensions learned from the similarity judgements, which predicts unseen human similarity judgements with high accuracy and offers an interpretable account of the mental representation of objects[52,57]. Here we used these object dimensions

to predict fMRI responses to object images. All data generation and processing methods are described in detail in the original data publication[57] and are only summarized here.

## Participants

The MRI dataset in the THINGS-data collection comprises data from three healthy volunteers (two female, one male; mean age, 25.33 years). The participants had normal or corrected-to-normal visual acuity and were right-handed. The behavioural dataset in the THINGS-data collection was obtained from 12,340 participants through the crowd-sourcing platform Amazon Mechanical Turk (6,619 female, 4,400 male, 56 other, 1,065 not reported; mean age, 36.71 years; s.d., 11.87 years; $n$ = 5,170 no age reported). The participants provided informed consent in participation and data sharing, and they received financial compensation for taking part in the studies. Data acquisition of the THINGS-data collection was approved by the National Institutes of Health Institutional Review Board (study protocol 93 M-0170, NCT00001360).

## Stimuli

All images were taken from the THINGS database[82]. The THINGS database contains 26,107 high-quality, coloured images of 1,854 object concepts from a wide range of nameable living and non-living objects, including non-countable substances (for example, 'grass'), faces (for example, 'baby', 'boy' and 'face') and body parts (for example, 'arm', 'leg' and 'shoulder'). The stimuli presented during fMRI included 720 object concepts from the THINGS database, with the first 12 examples of each concept selected for a total of 8,640 images. In addition, 100 of the remaining THINGS images were presented repeatedly in each session to estimate data reliability.

## Experimental procedure

Participants in the THINGS-fMRI experiment took part in 15–16 scanning sessions, with the first 1–2 sessions serving to acquire individual functional localizers for retinotopic visual areas and category-selective clusters (faces, body parts, scenes, words and objects). The main fMRI experiment comprised 12 sessions where participants were presented with the 11,040 THINGS images (8,740 unique images, catch trials excluded, 500 ms presentation followed by 4 s of fixation). For details on the procedure of the fMRI and behavioural experiments, please consult the original publication of the datasets[57].

Behavioural similarity judgements in the THINGS-data collection were collected in a triplet odd-one-out study using the online crowdsourcing platform Amazon Mechanical Turk. The participants were presented with three object images side by side and were asked to indicate which object they perceived to be the odd one out. Each task comprised 20 odd-one-out trials, and the participants could perform as many tasks as they liked.

## MRI data acquisition and preprocessing

Whole-brain fMRI images were acquired with 2 mm isotropic resolution and a repetition time of 1.5 s. The MRI data were preprocessed with the standard pipeline fMRIPrep[105], which included slice time correction, head motion correction, susceptibility distortion correction, co-registration between functional and T1-weighted anatomical images, brain tissue segmentation, and cortical surface reconstruction. Additionally, cortical flat maps were manually generated[106]. The fMRI data were denoised with a semi-automated procedure based on independent component analysis, which was developed specifically for the THINGS-fMRI dataset. The retinotopic mapping data and functional localizer data were used to define retinotopic visual regions as well as the category-selective regions used in this study. Image-wise response estimates were obtained by fitting a single-trial model to the fMRI time series of each functional run while accounting for variation in haemodynamic response shape and mitigating overfitting[107–109].

## Behavioural embedding

To predict the neural response to seen objects, we used a recent, openly available model of representational dimensions underlying human similarity judgements of objects[52]. This model was trained to estimate a low-dimensional, sparse and non-negative embedding predictive of individual trial choices in an odd-one-out task on 1,854 object images. The dimensions of this embedding have been demonstrated to be highly predictive of human similarity judgements while yielding human-interpretable dimensions reflecting both perceptual (for example, 'red' and 'round') and conceptual (for example, 'animal-related') object properties. We used a recent 66-dimensional embedding trained on 4.7 million odd-one-out judgements on triplets of 1,854 object images[57].

While the original embedding was trained on one example image for each of the 1,854 object concepts, it may not account for differences between exemplars of the same object concept. For example, the colour of the apple the model was trained on might have been red, while we also presented participants with images of a green apple. This may underestimate the model's potential to capture variance in brain responses to visually presented object images. To address this, we extended the original embedding by predicting the 66 object dimensions for each individual image in the THINGS database[82]. To this end, we used the neural network model CLIP-ViT, which is a multimodal model trained on image–text pairs and which was recently demonstrated to yield excellent prediction of human similarity judgements[65,69]. For each of the 1,854 object images, we extracted the activity pattern from the final layer of the image encoder. Then, for each of the 66 dimensions, we fitted a ridge regression model to predict dimension values, using cross-validation to determine the regularization hyperparameter. Finally, we applied the learned regression model to activations for all images in the THINGS database. This resulted is a 66-dimensional embedding that captures the mental representation of all 26,107 THINGS images. We used these predicted dimension values to predict fMRI responses to the subset of 8,740 unique images presented in fMRI, which yielded consistent improvements in explained variance for all dimensions (Supplementary Fig. 1).

## Encoding model

We used a voxel-wise encoding model of the 66-dimensional similarity embedding to predict image-wise fMRI responses to test (1) how well the model predicts neural responses in different parts of the visual system and (2) how neural tuning to individual dimensions maps onto the topography of visual cortex.

**Linear regression on fMRI single-trial estimates.** To test how well the core object dimensions predict brain responses in different parts of the visual system, we fit them to the fMRI single-trial response estimates using ordinary least squares regression. While most analyses in this work rely on a more powerful parametric modulation model estimated on time-series data (see below), we used single-trial responses for estimating the predictivity of the object dimensions, since this approach does not require extracting the contribution of the parametric modulators for estimating the explained variance of the general linear model. We evaluated the prediction performance of this encoding model in a leave-one-session-out cross-validation, using the average correlation between predicted and observed fMRI responses across folds. Within each cross-validation fold, we also computed a null distribution of correlation values based on 10,000 random permutations of the held-out test data. To assess statistical significance, we obtained voxel-wise $P$ values by comparing the estimated correlation with the generated null distribution and corrected for multiple comparisons on the basis of a false discovery rate of $P < 0.01$. We computed noise-ceiling-corrected $R^2$ values by dividing the original $R^2$ of the model by the noise ceiling estimates, for each voxel separately. These single-trial noise ceilings (Supplementary Fig. 7) were provided with the fMRI dataset and were

computed on the basis of estimates of the signal and noise variance, which were based on the variability of responses to repeatedly presented images[57].

**Parametric modulation on fMRI time series.** To evaluate the contribution of individual object dimensions to the neural response in a given voxel, we used a parametric modulation model on the voxel-wise time-series data. In this parametric modulation, a general onset regressor accounts for the average response across all trials, and a set of 66 parametric modulators account for the modulation of the BOLD signal by individual object dimensions. To compile the parametric modulation model, we constructed dimension-specific onset regressors and mean-centred each parametric modulator to make them orthogonal to the general onset regressor. We then convolved these regressors with a haemodynamic response function (HRF) to obtain predictors of the BOLD response. To account for variation in the shape of the HRF, we determined the best-fitting HRF for each voxel on the basis of a library of 20 HRFs[107,108]. The resulting design matrix was then concatenated and fit to the fMRI time-series data. To mitigate overfitting, we regularized the regression weights using fractional ridge regression[109]. We chose a range of regularization parameters from 0.10 to 0.90 in steps of 0.10 and from 0.90 to 1.00 in steps of 0.01 to more densely sample values that reflect less regularization. We determined the best hyperparameter combination (20 HRFs and 26 regularization parameters) for each voxel on the basis of the amount of variance explained in a 12-fold between-session cross-validation. Finally, we fit the model with the best hyperparameter combination per voxel to the entire dataset, yielding 66 statistical maps of regression weights representing the voxel-wise contribution of individual object dimensions in predicting the fMRI signal. The regularization hyperparameter turned out to be small throughout visual cortex (Supplementary Fig. 8), demonstrating that the regularization of regression weights had little impact on the absolute size of regression weights. While our analysis was focused on individual participants, we also estimated the consistency of the tuning maps of individual dimensions across participants. To this end, we used a number of individually defined regions of interest as anchor points for quantifying similarities and differences between these maps. First, for each dimension separately, we obtained mean $\beta$ patterns across these regions, including early visual retinotopic areas (V1–V3 and hV4) as well as face- (FFA and OFA), body- (EBA) and scene-selective (PPA, OPA and MPA) regions. Face-, body- and scene-selective regions were analysed separately for each hemisphere to account for potential lateralized effects, and voxels with a noise ceiling smaller than 2% were excluded from the analysis. Finally, to quantify the replicability across participants, we computed the inter-participant correlation on the basis of these mean $\beta$ patterns, separately for each dimension (Supplementary Fig. 4).

**Regional tuning profiles and most representative object images**

To explore the functional selectivity implied by regional tuning to core object dimensions, we extracted tuning profiles for different visual brain regions and related them to the multidimensional representation of all object images in the THINGS database[82] using a high-throughput approach. First, we extracted the regression weights resulting from the parametric modulation model in different visual brain regions: V1, V2, V3, hV4, OFA, FFA, EBA, PPA, MPA and OPA. We then averaged these regional tuning profiles across participants and set negative weights to zero, given that the predicted dimensions reflect non-negative values as well. We plotted the regional tuning profiles as rose plots to visualize the representation of core object dimensions in these brain regions. To explore the regional selectivity for specific object images, we determined the cosine similarity between each regional tuning profile and the model representation of all 26,107 images in the THINGS database. This allowed us to identify those THINGS images that are

most representative of the local representational profile in different visual brain regions.

**Representational sparseness**

We estimated the sparseness of the representation of core object dimensions on the basis of the regression weights from the parametric modulation model. Given our aim of identifying local clusters of similarly tuned voxels, we performed spatial smoothing on the regression weight maps (4 mm full-width at half-maximum) to increase the spatial signal-to-noise ratio. We then took the vectors representing the 66-dimensional tuning profile for each voxel and removed negative vector elements, mirroring the analysis of the regional tuning profiles. We computed the sparseness of the resulting voxel-wise tuning vectors on the basis of a previously introduced sparseness measure, which is based on the normalized relationship between the L-1 and L-2 norm of a vector[110]:

$$s(\mathbf{x}) = \frac{\sqrt{n} - \sum |x_i| / \sqrt{\sum x_i^2}}{\sqrt{n} - 1}$$

where $s$ indicates the sparseness of the $n$-dimensional input vector $\mathbf{x}$. A sparseness value of 1 indicates a perfectly sparse representation where all vector elements except one have the same value. In turn, a value of 0 indicates a perfectly dense representation where all elements have identical values. We computed this sparseness measure over the regression weights in each voxel, which yielded a sparseness measure as a single value per voxel. To assess their statistical significance, we first identified the distribution of sparseness values in a noise pool of voxels. This noise pool included voxels where the parametric modulation model predicted the fMRI signal poorly in the cross-validation procedure ($R^2 < 0.0001$). Since visual inspection of sparseness histograms suggested a log-normal distribution, we log-transformed all sparseness values to convert them to a normal distribution. Finally, we estimated the mean and standard deviation of the sparseness distribution in the noise pool, allowing us to obtain $z$ and $P$ values of the sparseness in each voxel.

On the basis of these results, we explored whether local clusters of representational sparseness are indicative of brain regions with high functional selectivity. To this end, we identified two regional clusters of high sparseness values which were present in all participants and which had not yet been defined on the basis of the functional localizer experiment (see 'MRI data acquisition and preprocessing'). On the basis of visual inspection of the sparseness maps, we defined two regions of interest. The first region of interest was located in the right ventro-temporal cortex, anterior to anatomically defined area FG4 (ref. 88) and functionally defined FFA, but posterior to the anterior temporal face patch[89]. The second region of interest was located in the orbitofrontal cortex. We probed the functional selectivity of these sparsely tuned regions by extracting regional tuning profiles and determining the most representative object images as described in the previous section.

**Variance partitioning of object-category-based versus dimension-based models**

The aim of the variance partitioning was to test whether object dimensions or object categories offer a better model of neural responses to object images. To this end, we compiled a multidimensional and categorical model and compared the respective amount of shared and unique variance explained by these models.

We used 50 superordinate object categories provided in the THINGSplus metadata collection to construct a category encoding model[97] (see Supplementary Methods 3 for a full list). To account for cases where images contained multiple objects (for example, an image of 'ring' might also contain a finger), we used the image annotations in the THINGSplus metadata[97] and manually matched these annotations to objects in the THINGS database for all images presented in the fMRI

experiment. Lastly, we added two more categories by manually identifying images containing human faces or body parts. We then compiled an encoding model with 52 binary regressors encoding the high-level categories of all respective objects.

Next, we compiled a corresponding encoding model of object dimensions. Note that we predicted that this model would outperform the categorical model in explaining variance in neural responses. To conservatively test this prediction, we biased our analysis in favour of the categorical model by selecting fewer dimensions than categories. To this end, for each category we identified the object dimension with the strongest relationship based on the area under the curve metric. Since some dimensions are diagnostic for multiple categories (for example, 'animal-related' might be the most diagnostic dimension for both 'bird' and 'insect'), this resulted in a one-to-many mapping between 30 dimensions and 50 categories (see Supplementary Methods 3 for a full list of selected dimensions).

To compare the predictive potential of these two models, we fitted them to the fMRI single-trial responses in a voxel-wise linear regression and performed variance partitioning. To estimate the uniquely explained variance, we first orthogonalized the target model and the data with respect to the other model[111]. This effectively removed the shared variance from both the target model and the data. We then fit the residuals of the target model to the residuals of the data and calculated the coefficient of determination ($R^2$) in a 12-fold between-session cross-validation as an estimate of the unique variance explained by the target model. We then estimated the overall variance explained by both models by concatenating the two models, fitting the resulting combined model to the data and determining the cross-validated $R^2$ estimate. Lastly, we computed an estimate of the shared variance explained by the two models by subtracting the uniquely explained variances from the overall explained variance. For visualization purposes, $R^2$ values were normalized by the noise ceiling estimates provided with the fMRI dataset[57] (Supplementary Fig. 7). We also visualized the relationship between the performance of both models quantitatively. To that end, we selected voxels with a noise ceiling of greater than 5% in early (V1–V3) and higher-level (face-, body- and scene-selective) regions of interest and created scatter plots comparing the variance uniquely explained by the category- and dimensions-based models in these voxels (Supplementary Fig. 5). To summarize the extent of explained variance, we computed median and maximum values for the shared and unique explained variances in these voxels.

### Reporting summary

Further information on research design is available in the Nature Portfolio Reporting Summary linked to this article.

## Data availability

The data supporting our analyses were obtained from the publicly available THINGS-fMRI dataset. The fMRI dataset is accessible on Open-Neuro (https://doi.org/10.18112/openneuro.ds004192.v1.0.5) and via Figshare at https://doi.org/10.25452/figshare.plus.c.6161151 (ref. 112). The object dimensions embedding underlying behavioural similarity judgements that was used to predict the fMRI responses is available at the Open Science Framework repository (https://osf.io/f5rn6/). The higher-level object category labels that were used to construct a categorical model of object responses are part of the THINGSplus metadata and available at the Open Science Framework (https://osf.io/jum2f/). The BOLD5000 fMRI data, including all image stimuli, are openly available on the KiltHub repository hosted on Figshare at https://doi.org/10.1184/R1/14456124 (ref. 113).

## Code availability

The Python code (version 3.7.6) used for data processing, analysis and visualization in this study is publicly available on GitHub (https://github.com/ViCCo-Group/dimension_encoding).

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

## Acknowledgements

We thank P. Kaniuth for his help with image-wise dimension predictions, M. Holzner for her help with identifying background objects in images and finding copyright-free alternative images for publication, and J. Prince for sharing cortical flat maps for the BOLD5000 data. This work was supported by a doctoral student fellowship awarded to O.C. by the Max Planck School of Cognition, the Intramural Research Program of the National Institutes of Health (ZIA-MH-002909), under National Institute of Mental Health Clinical Study Protocol 93-M-1070 (NCT00001360), a research group grant by the Max Planck Society awarded to M.N.H., the ERC Starting Grant project COREDIM (ERC-StG-2021-101039712) and the Hessian Ministry of Higher Education, Science, Research and Art (LOEWE Start Professorship to M.N.H. and Excellence Program 'The Adaptive Mind'). The funders had no role in study design, data collection and analysis, decision to publish or preparation of the manuscript.

## Author contributions

O.C., C.I.B. and M.N.H. conceived the study. O.C. carried out the data analysis and wrote the original draft of the manuscript. C.I.B. and M.N.H. reviewed the manuscript and provided critical feedback. M.N.H. supervised the project.

## Funding

## Competing interests

The authors declare no competing interests.

## Additional information

**Extended data** is available for this paper at https://doi.org/10.1038/s41562-024-01980-y.

**Correspondence and requests for materials** should be addressed to Oliver Contier.

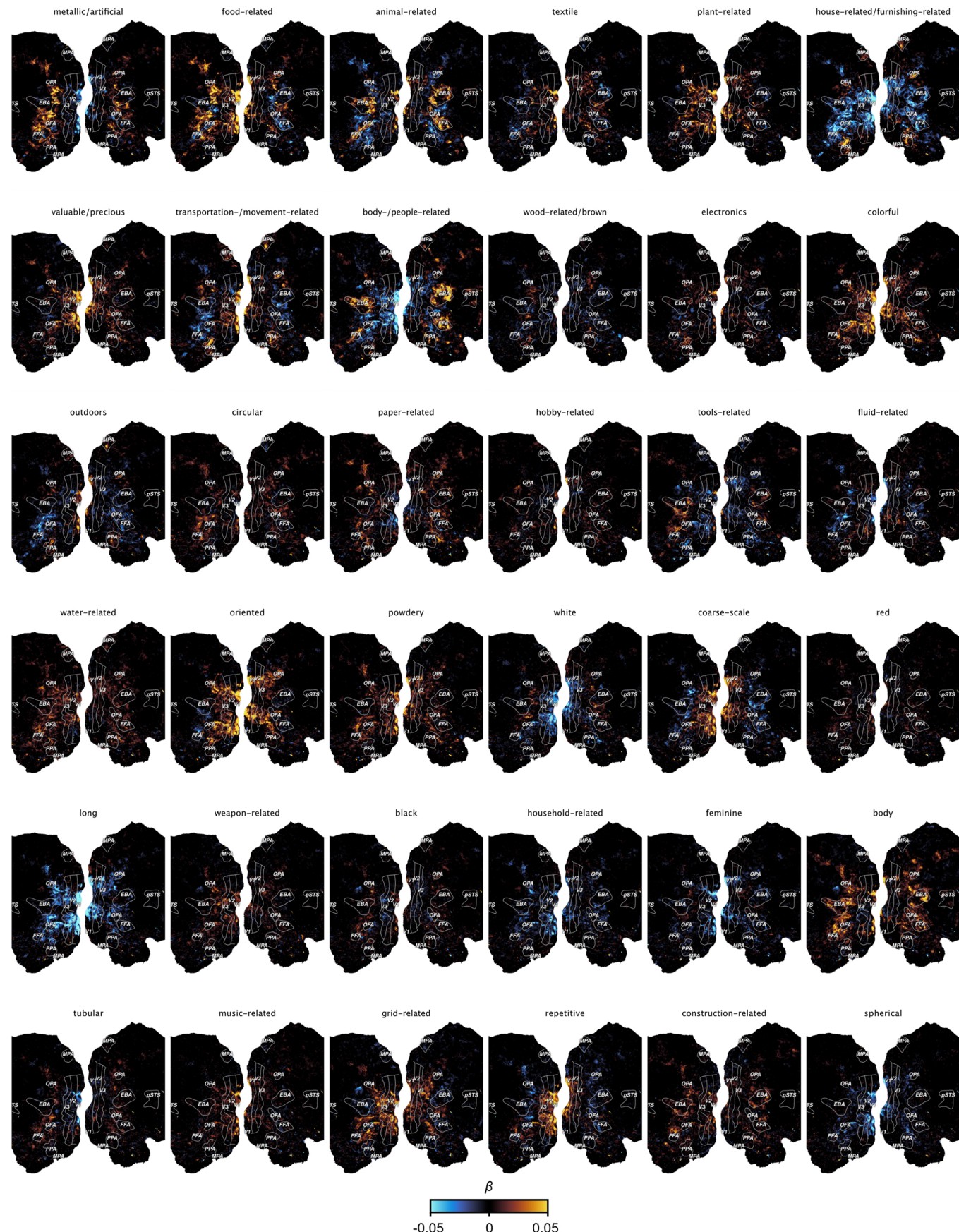

**Extended Data Fig. 1 | Dimension tuning maps 1-36 for Participant 1.** Colors indicate regression weights for each dimension predictor from the parametric modulation encoding model. Labels indicate regions of interest on the cortex: V1-V3: primary - tertiary visual cortex, OFA: occipital face area, FFA: fusiform face area, EBA: extrastriate body area, PPA: parahippocampal place area, MPA: medial place area, OPA: occipital place area.

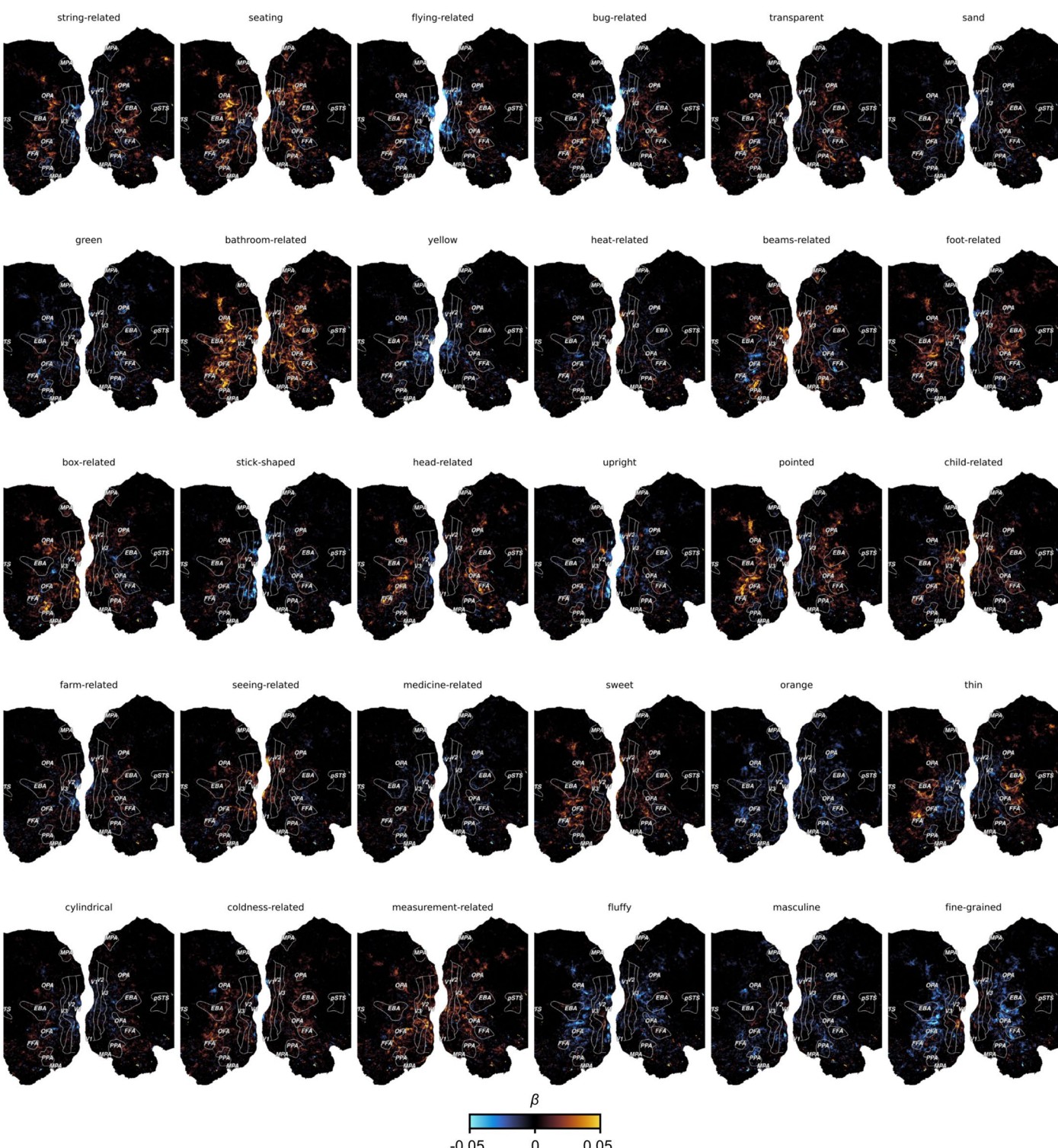

**Extended Data Fig. 2 | Dimension tuning maps 37-66 for Participant 1.** Colors indicate regression weights for each dimension predictor from the parametric modulation encoding model. Labels indicate regions of interest on the cortex: V1-V3: primary - tertiary visual cortex, OFA: occipital face area, FFA: fusiform face area, EBA: extrastriate body area, PPA: parahippocampal place area, MPA: medial place area, OPA: occipital place area.

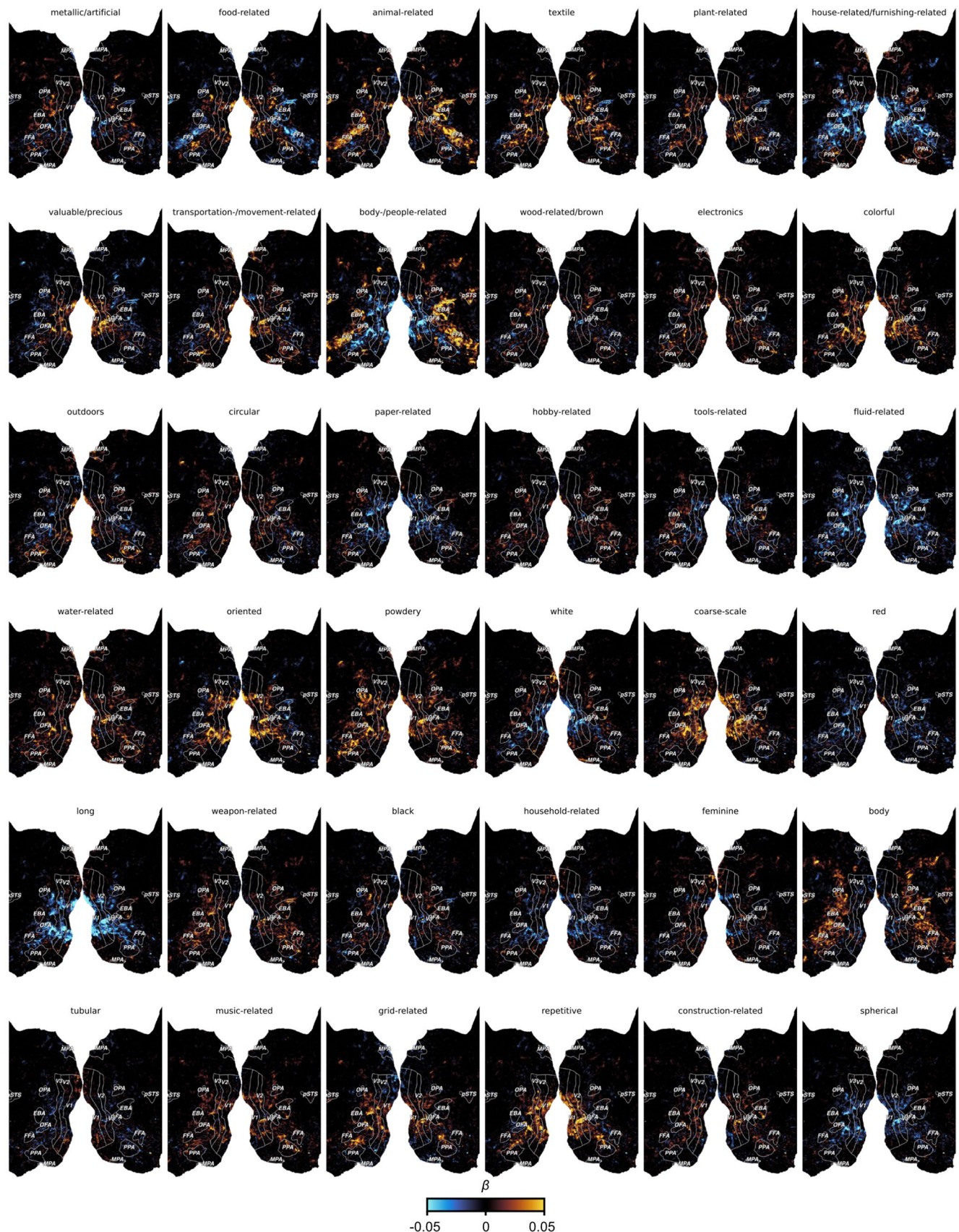

**Extended Data Fig. 3 | Dimension tuning maps 1-36 for Participant 2.** Colors indicate regression weights for each dimension predictor from the parametric modulation encoding model. Labels indicate regions of interest on the cortex: V1-V3: primary - tertiary visual cortex, OFA: occipital face area, FFA: fusiform face area, EBA: extrastriate body area, PPA: parahippocampal place area, MPA: medial place area, OPA: occipital place area.

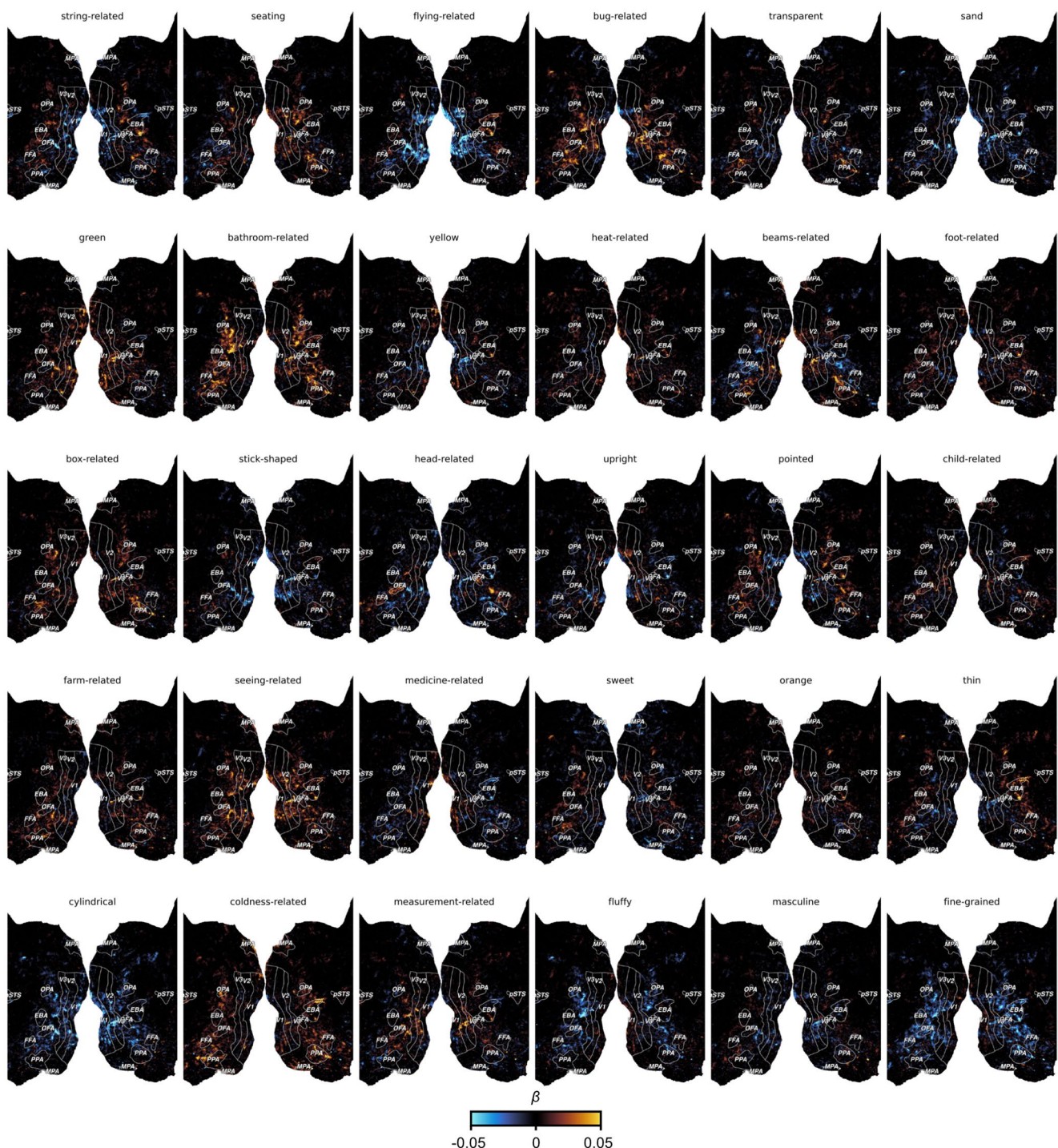

**Extended Data Fig. 4 | Dimension tuning maps 37-66 for Participant 2.** Colors indicate regression weights for each dimension predictor from the parametric modulation encoding model. Labels indicate regions of interest on the cortex: V1-V3: primary - tertiary visual cortex, OFA: occipital face area, FFA: fusiform face area, EBA: extrastriate body area, PPA: parahippocampal place area, MPA: medial place area, OPA: occipital place area.

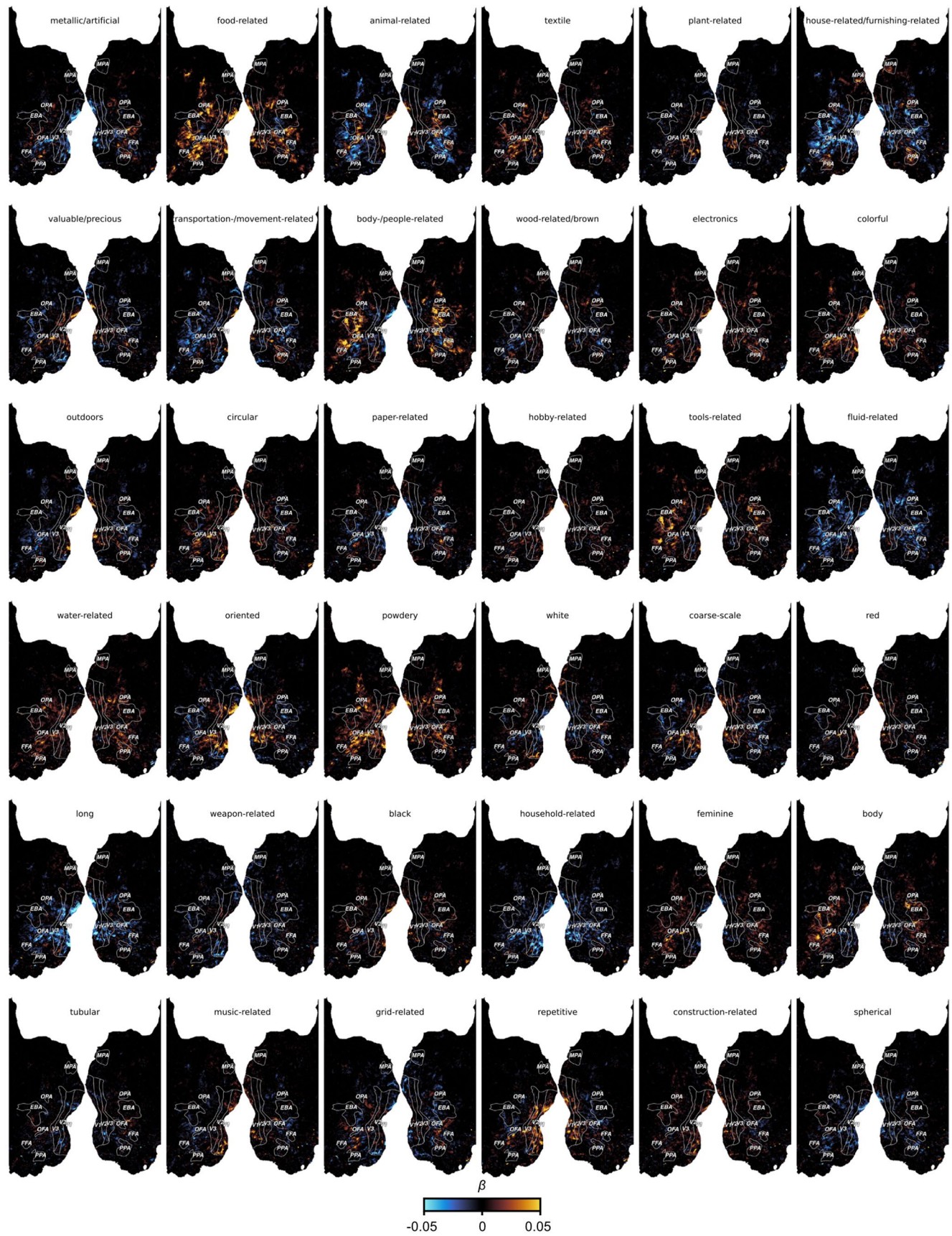

**Extended Data Fig. 5 | Dimension tuning maps 1-36 for Participant 3.** Colors indicate regression weights for each dimension predictor from the parametric modulation encoding model. Labels indicate regions of interest on the cortex: V1-V3: primary - tertiary visual cortex, OFA: occipital face area, FFA: fusiform face area, EBA: extrastriate body area, PPA: parahippocampal place area, MPA: medial place area, OPA: occipital place area.

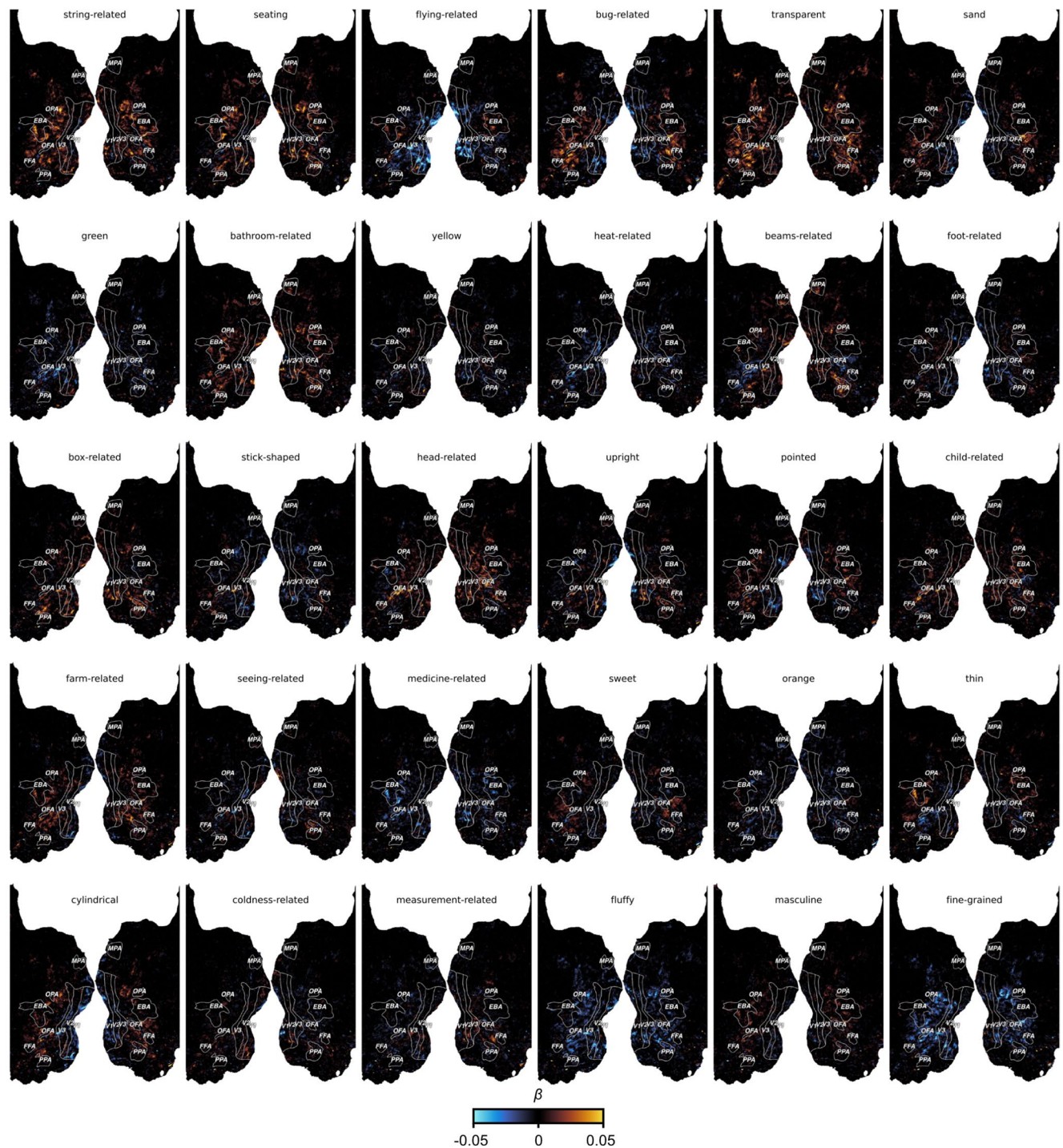

**Extended Data Fig. 6 | Dimension tuning maps 37-66 for Participant 3.** Colors indicate regression weights for each dimension predictor from the parametric modulation encoding model. Labels indicate regions of interest on the cortex: V1-V3: primary - tertiary visual cortex, OFA: occipital face area, FFA: fusiform face area, EBA: extrastriate body area, PPA: parahippocampal place area, MPA: medial place area, OPA: occipital place area.

# Reporting Summary

## Statistics

For all statistical analyses, confirm that the following items are present in the figure legend, table legend, main text, or Methods section.

| n/a | Confirmed | |
|---|---|---|
| ☐ | ☒ | The exact sample size (*n*) for each experimental group/condition, given as a discrete number and unit of measurement |
| ☐ | ☒ | A statement on whether measurements were taken from distinct samples or whether the same sample was measured repeatedly |
| ☐ | ☒ | The statistical test(s) used AND whether they are one- or two-sided<br>*Only common tests should be described solely by name; describe more complex techniques in the Methods section.* |
| ☒ | ☐ | A description of all covariates tested |
| ☒ | ☐ | A description of any assumptions or corrections, such as tests of normality and adjustment for multiple comparisons |
| ☐ | ☒ | A full description of the statistical parameters including central tendency (e.g. means) or other basic estimates (e.g. regression coefficient) AND variation (e.g. standard deviation) or associated estimates of uncertainty (e.g. confidence intervals) |
| ☐ | ☒ | For null hypothesis testing, the test statistic (e.g. *F*, *t*, *r*) with confidence intervals, effect sizes, degrees of freedom and *P* value noted<br>*Give P values as exact values whenever suitable.* |
| ☒ | ☐ | For Bayesian analysis, information on the choice of priors and Markov chain Monte Carlo settings |
| ☒ | ☐ | For hierarchical and complex designs, identification of the appropriate level for tests and full reporting of outcomes |
| ☐ | ☒ | Estimates of effect sizes (e.g. Cohen's *d*, Pearson's *r*), indicating how they were calculated |

*Our web collection on statistics for biologists contains articles on many of the points above.*

## Software and code

Policy information about availability of computer code

| Data collection | No additional data was collected for this manuscript. For a full description of the data acquisition including relevant computer software, see Hebart et al. 2023, https://doi.org/10.7554/eLife.82580 |
|---|---|
| Data analysis | python (3.7.6) custom code with specified dependencies available at https://github.com/ViCCo-Group/dimension_encoding/ |

For manuscripts utilizing custom algorithms or software that are central to the research but not yet described in published literature, software must be made available to editors and reviewers. We strongly encourage code deposition in a community repository (e.g. GitHub). See the Nature Portfolio guidelines for submitting code & software for further information.

## Data

Policy information about availability of data

All manuscripts must include a data availability statement. This statement should provide the following information, where applicable:
- Accession codes, unique identifiers, or web links for publicly available datasets
- A description of any restrictions on data availability
- For clinical datasets or third party data, please ensure that the statement adheres to our policy

The data supporting our analyses were obtained from the publicly available THINGS-fMRI dataset. The fMRI dataset is accessible on OpenNeuro (https://doi.org/10.18112/openneuro.ds004192.v1.0.5) and Figshare (https://doi.org/10.25452/figshare.plus.c.6161151). The object dimensions embedding underlying behavioral similarity judgements which was used to predict the fMRI responses is available at the Open Science Framework repository (https://osf.io/f5rn6/). The

higher-level object category labels which were used to construct a categorical model of object responses are part of the THINGSplus metadata and available at the Open Science Framework (https://osf.io/jum2f/). The BOLD 5000 data, including all images e.g. from the SUN database are openly available on figshare (https://doi.org/10.1184/R1/14456124).

## Research involving human participants, their data, or biological material

Policy information about studies with [underline]human participants or human data[/underline]. See also policy information about [underline]sex, gender (identity/presentation), and sexual orientation[/underline] and [underline]race, ethnicity and racism[/underline].

| | |
|---|---|
| Reporting on sex and gender | This study used already openly available data. No additional participants were recruited. More details can be found in the manuscript describing the data generation methods and consent information (https://elifesciences.org/articles/82580#s4). <br><br> 2 of the 3 participants self-reported female gender. Neither sex nor gender was considered in study design. Neither sex- nor gender-related analyses were performed because the data, due to the small sample size, is unsuited for studying inter-individual effects. Participants had given consent for obtaining and sharing individual-level data. |
| Reporting on race, ethnicity, or other socially relevant groupings | No other socially relevant categorization variables were used in this manuscript. |
| Population characteristics | All participants were asked to report their age (Mean age at beginning of study: 25.33 years). |
| Recruitment | This study used already openly available data. No additional participants were recruited. |
| Ethics oversight | n/a |

Note that full information on the approval of the study protocol must also be provided in the manuscript.

# Field-specific reporting

Please select the one below that is the best fit for your research. If you are not sure, read the appropriate sections before making your selection.

☒ Life sciences ☐ Behavioural & social sciences ☐ Ecological, evolutionary & environmental sciences

For a reference copy of the document with all sections, see [underline]nature.com/documents/nr-reporting-summary-flat.pdf[/underline]

# Life sciences study design

All studies must disclose on these points even when the disclosure is negative.

| | |
|---|---|
| Sample size | Analysis was performed on three subjects individually. The number of subjects in the open dataset we used is limited by the feasibility of data acquisition, which focused on densely sampled, large-scale recordings of neural responses for each individual subject instead of sampling a larger population. |
| Data exclusions | None of the THINGS-fMRI data had been excluded for this work. In the BOLD 5000 reanalysis, we excluded trials showing images from the SUN database because they did not contain objects. |
| Replication | We replicated our results in an independent dataset (BOLD5000), based on three different participants and different sets of stimuli (ImageNet and MS CoCo). All attempts at replication were successfull. |
| Randomization | Randomization did not apply to this work since we did not experimentally manipulate any variables. Instead, we reanalyzed already existing data. |
| Blinding | Blinding is not applicable to this work since we did not experimentally manipulate any variables. |

# Reporting for specific materials, systems and methods

We require information from authors about some types of materials, experimental systems and methods used in many studies. Here, indicate whether each material, system or method listed is relevant to your study. If you are not sure if a list item applies to your research, read the appropriate section before selecting a response.

## Materials & experimental systems

| n/a | Involved in the study |
|---|---|
| ☒ | ☐ Antibodies |
| ☒ | ☐ Eukaryotic cell lines |
| ☒ | ☐ Palaeontology and archaeology |
| ☒ | ☐ Animals and other organisms |
| ☒ | ☐ Clinical data |
| ☒ | ☐ Dual use research of concern |
| ☒ | ☐ Plants |

## Methods

| n/a | Involved in the study |
|---|---|
| ☒ | ☐ ChIP-seq |
| ☒ | ☐ Flow cytometry |
| ☐ | ☒ MRI-based neuroimaging |

## Plants

| | |
|---|---|
| Seed stocks | *Report on the source of all seed stocks or other plant material used. If applicable, state the seed stock centre and catalogue number. If plant specimens were collected from the field, describe the collection location, date and sampling procedures.* |
| Novel plant genotypes | *Describe the methods by which all novel plant genotypes were produced. This includes those generated by transgenic approaches, gene editing, chemical/radiation-based mutagenesis and hybridization. For transgenic lines, describe the transformation method, the number of independent lines analyzed and the generation upon which experiments were performed. For gene-edited lines, describe the editor used, the endogenous sequence targeted for editing, the targeting guide RNA sequence (if applicable) and how the editor was applied.* |
| Authentication | *Describe any authentication procedures for each seed stock used or novel genotype generated. Describe any experiments used to assess the effect of a mutation and, where applicable, how potential secondary effects (e.g. second site T-DNA insertions, mosiacism, off-target gene editing) were examined.* |

## Magnetic resonance imaging

### Experimental design

| | |
|---|---|
| Design type | Event-related task fMRI. |
| Design specifications | 11,040 images (8,740 unique images, catch trials excluded, 500 ms presentation followed by 4 s of fixation). For details on the procedure of the fMRI and behavioral experiments, please consult the original publication of the dataset (https://elifesciences.org/articles/82580) |
| Behavioral performance measures | Participants responded to catch trials in order to stay engaged. Response accuracy was and catch trials were not analyzed. |

### Acquisition

| | |
|---|---|
| Imaging type(s) | functional |
| Field strength | 3 |
| Sequence & imaging parameters | Gradient echo EPI, 2 mm isometric resolution, FOV = 192 mm × 192 mm, matrix size = 96 × 96; slice thickness: 2 mm, axial orientation, TR/TE/flip angle = 1.5s/33ms/75° |
| Area of acquisition | whole-brain |
| Diffusion MRI | ☐ Used    ☒ Not used |

### Preprocessing

| | |
|---|---|
| Preprocessing software | The data used in this publication was already provided in preprocessed form. Additional smoothing (fwhm=4mm) was only performed for the sparseness analysis using the nilearn python library. |
| Normalization | Data were not normalized. |
| Normalization template | n/a |
| Noise and artifact removal | None |
| Volume censoring | None |

## Statistical modeling & inference

| | |
|---|---|
| Model type and settings | Voxel-wise encoding model involving a cross-validated train-test procedure. |
| Effect(s) tested | Variance explained (r-squared) of the entire model. |

Specify type of analysis: ☐ Whole brain ☐ ROI-based ☒ Both

Anatomical location(s): Object category-selective clusters were determined based on a standard functional localizer experiment. Similarly, retinotopic visual regions were determined based on a population receptive field experiment.

| | |
|---|---|
| Statistic type for inference<br>(See Eklund et al. 2016) | voxel-wise |
| Correction | FDR |

## Models & analysis

| n/a | Involved in the study |
|---|---|
| ☒ | ☐ Functional and/or effective connectivity |
| ☒ | ☐ Graph analysis |
| ☐ | ☒ Multivariate modeling or predictive analysis |

Multivariate modeling and predictive analysis: Independent variables: Object dimensions. Dependent variables: Voxel-wise responses to each object image. Average prediction performance was evaluated with a leave-one-session-out cross-validation and statistical significance was tested via permutation test (10,000 random permutations in each cross-validation fold, FDR p<0.01).

