## [Peer Review File · Nature Human Behaviour]

Peer Review Information

Journal: Nature Human Behaviour

Manuscript Title: Distributed representations of behavior-derived object dimensions in the human visual system

Corresponding author name(s): Oliver Contier

Reviewer Comments & Decisions:

Decision Letter, initial version:

19th January 2024

Dear Mr Contier,

Thank you once again for your manuscript, entitled "Distributed representations of behaviorally-relevant object dimensions in the human visual system", and for your patience during the peer review process.

Your Article has now been evaluated by 3 referees. You will see from their comments copied below that, although they find your work very interesting, they have raised quite substantial concerns. In light of these comments, we cannot accept the manuscript for publication, but would be interested in considering a revised version if you are willing and able to fully address reviewer and editorial concerns.

We hope you will find the referees' comments useful as you decide how to proceed. If you wish to submit a substantially revised manuscript, please bear in mind that we will be reluctant to approach the referees again in the absence of major revisions. We are committed to providing a fair and constructive peer-review process. Do not hesitate to contact us if there are specific requests from the reviewers that you believe are technically impossible or unlikely to yield a meaningful outcome.

In particular, we think that the following issues will be most important to address:

- The question of whether your results would replicate in other datasets (Ref #1). While we recognize that it is unlikely that there exist other datasets using the same range of stimuli to allow direct replications, it would be beneficial to provide conceptual replications if possible.
- The question of how much variance is explained by your model (Ref #2) and the need for a direct comparison with other models including category-based and shape-based ones (Ref #2 and #3).

If you wish to submit a suitably revised manuscript, we would hope to receive it within 4 months. I would be grateful if you could contact us as soon as possible if you foresee difficulties with meeting this target resubmission date.

- Include a "Response to the editors and reviewers" document detailing, point-by-point, how you addressed each editor and referee comment. If no action was taken to address a point, you must provide a compelling argument. When formatting this document, please respond to each reviewer comment individually, including the full text of the reviewer comment verbatim followed by your response to the individual point. This response will be used by the editors to evaluate your revision and sent back to the reviewers along with the revised manuscript.
- Highlight all changes made to your manuscript or provide us with a version that tracks changes.

[REDACTED]

Thank you for the opportunity to review your work. Please do not hesitate to contact me if you have any questions or would like to discuss the required revisions further.

Sincerely,
[REDACTED]

REVIEWER COMMENTS:

Reviewer #1:

Remarks to the Author:

This work presents a comprehensive study on the distributed representations of behaviorally relevant object dimensions in the human visual system. The study challenges the traditional category-centric framework of object vision, which posits a hierarchical processing of increasingly complex image features for object recognition and categorization. This conventional view is limited as it does not fully encompass the diverse behavioral goals supported by object vision.

The authors propose an alternative framework, where the visual cortex is organized based on continuous dimensions reflecting more general object properties that were extracted from behavioral data. This perspective accounts for the rich meaning and behavioral relevance of objects beyond discrete labels, incorporating various visual and semantic properties. The study utilizes a large-scale analysis of human similarity judgments to map these behaviorally relevant dimensions onto the brain, revealing broadly distributed representations across the visual cortex. These representations demonstrate selectivity to a wide variety of novel dimensions while capturing known selectivities for visual features and categories. This approach reveals mixed selectivity in much of the visual cortex and sparse selectivity in category-selective clusters.

The methodology involves modeling neural representation of objects by using the THINGS-data collection, which includes fMRI data for thousands of naturalistic object images and millions of behavioral similarity judgments. The study identifies 66 core dimensions from this data, which are used to predict brain responses to objects. These dimensions capture external behavior such as categorization and typicality judgments, underscoring their behavioral relevance. The findings show that these core object dimensions are reflected in widespread fMRI activity patterns throughout the visual system, and encoding based on these dimensions significantly outperforms category-based approaches. The study also challenges previous notions that perceived similarity information is confined primarily to higher-level visual cortex, and demonstrates that behaviorally relevant information about objects is distributed throughout the visual processing hierarchy, including early cortical processing stages.

In terms of functional selectivity, the study finds that category-selective brain regions are sparsely tuned to fewer behaviorally relevant object dimensions, indicating that these regions respond primarily to a small subset of dimensions. In contrast, non-category-selective regions exhibit a denser representation, responding broadly to diverse dimensions. This suggests that category-selectivity may be a special case of sparse tuning within a broader set of distributed dimension tuning maps. The study uses this sparseness principle to identify two novel sparsely tuned regions of the brain, which are tuned to desserts and animal-faces. This finding illustrates the impact of this new dimensional representational paradigm and makes clear the core contribution of the work.

The discussion emphasizes the importance of considering the entire visual system for identifying the behavioral relevance of visual responses. The study advances a framework that integrates regional

specialization within a broader multidimensional topography, providing a more comprehensive view of object representations. The study acknowledges its limitations, such as not identifying behaviorally-relevant dimensions specific to each individual participant and excluding non-object images like text from the analysis. Future research could extend these insights by exploring additional image classes and behaviors.

Strengths:

Innovative Framework: The paper presents an innovative approach to understanding object vision, moving beyond the traditional category-centric model. It introduces a novel framework that considers continuous dimensions of object properties, offering a more comprehensive view of how the visual system processes behaviorally relevant information and structures perceptual conscious states around a rich multidimensional latent space.

Comprehensive Data Analysis: The use of the THINGS-data collection, encompassing thousands of object images and millions of behavioral similarity judgments, provides a robust and diverse dataset. This extensive data collection enhances the credibility and depth of the study's findings. The methodology for the analysis, which includes the identification of 66 core object dimensions and their mapping onto brain activity patterns using fMRI, is both rigorous and advanced. The performance of the new encoding model demonstrates a high level of technical competence.

Insightful Findings: The study offers significant insights into the distributed representations of object information in the visual cortex. By showing that these representations span the entire visual processing hierarchy, the paper challenges previous beliefs about the confinement of perceived similarity information to higher-level visual cortex. The paper also applies their methodology to identify two completely novel sparsely tuned brain regions.

Clarity in Results and Implications: The results are clearly presented and logically lead to the conclusion that a multidimensional model offers more explanatory power than a category-centric approach.

Potential for Broader Impact: The findings have implications for a wide range of fields, including neuroscience, psychology, and artificial intelligence. The paper's multidimensional approach to object vision could influence future research and applications in these areas.

Weaknesses:

The authors do a good job of identifying the limitations of their work and identifying future areas of research, some notable ones the authors do not clearly address are outlined below.

Generalizability of Results: The study's findings are specific to a single dataset and behavioral methodology, with the generalizability of its findings still unknown. Future research could benefit from validating these results across different populations and using varied datasets to ensure broader applicability.

CLIP Pretraining Biases: The use of the CLIP-VIT model to encode images into the 66-dimensional behavioral embedding might introduce additional biases in the embedding due to the language/image pretraining of the model that are not explored in this work. The authors also don't perform any validation against training data leakage, as it is possible some of the images in the THINGS dataset were part of the training dataset for CLIP, and might have polluted some of the results.

In conclusion, this paper represents a significant contribution to our understanding of the human visual system. Its strengths far outweigh its weaknesses, offering a fresh perspective on object vision. The innovative approach, comprehensive data analysis, and insightful findings make it a valuable addition to the field. The weaknesses identified provide opportunities for further research and refinement, which could enhance the impact and applicability of this important work. Overall, I fully recommend this paper be accepted with no hesitations.

Reviewer #2:

Remarks to the Author:

The manuscript by Contier et al. tackles the much-researched but still un-answered question about the nature of the organization of visual representations in cortex. The current dominant view is one of a

"simple-to-complex" hierarchy starting in primary visual cortex, where neurons show orientation selectivity, and then progressing down the ventral visual stream to more and more complex selectivities, to selectivity for complex object classes such as faces (in the FFA), body parts (EBA), words (VWFA), and places (PPA) and the like, up to concept selectivity in the anterior temporal lobe. Yet, while a number of previous studies have posited principles by which the selectivities for complex objects could be arranged across cortex, none of these proposals have been compelling. Part of the reason is that prior approaches either focused on simple dimensions (such as size or animacy) with insufficient representational capacity as general models of object representations, or the studies used a large set of objects and then used a data-driven approach to discern the axes along which responses varied across cortex (determined, e.g., through principal component analysis; see, for example, Huth et al., *Nature*, 2016), with the shortcoming of this latter approach being that the resulting axes generally do not easily correspond to meaningful distinctions apparent to human observers (see, e.g., Barsalou, *Neuropsychologia*, 2017). In contrast, the present manuscript builds on the very interesting prior behavioral work by the two senior authors, who had analyzed a vast behavioral data set to isolate a set of dimensions that describe human similarity judgments for a large object database (the THINGS database). The objective of the current manuscript was to analyze fMRI responses to a (large) subset of these objects and to see whether the behavioral dimensions were useful as a representational space also for the objects' neural representations across visual cortex. If successful, this would present a major advance in our understanding of the organization of object representations in cortex.

To this end, the authors fit a voxel-wise encoding model of objects based on their 66 "core object dimensions" to the neural data, and then explored how well that model was able to predict responses to held-out objects that were not in the training set. Yet, this question of model fit is a major concern with the manuscript: The main figures in the manuscript focus on the "noise ceiling corrected R^2 ". This raises the critical question of what the noise ceiling is: what is the denominator used to normalize the correlation/explained variance values? Surprisingly, this information is missing from the manuscript. Supplementary Figure 2 appears to show a maximum correlation of 0.2 of predicted and held-out data (without noise ceiling normalization), corresponding to a maximum explained variance of only 4%, with most values appearing considerably lower. The authors need to provide this information on the noise ceiling to make it possible to assess model fit, as they did in their 2023 *eLife* paper. Supplementary Figure 2 suggests a rather low predictive power of their scheme, both absolutely and also relatively to Huth et al., which found that semantic axes obtained from the responses of all but one subject were able to account (in the held-out subject) for about 10 times the variance that Contier et al.'s model can explain. Granted, Huth et al.'s representation was data-driven, but the fact remains that the explained variance by Contier et al. appears quite minuscule. The authors stress that their analyses reveal "distributed tuning maps" (l. 374), with activity patterns "mirroring" behaviorally-relevant information. However, even apart from the low % variance explained (that would suggest use of a more cautious term than "mirroring"), the "maps" just seems like a lot of blobs, with little apparent structure.

A core claim of the manuscript is that the study results suggest a "behavior-centric view on visual processing in the human brain." Yet, to make this statement non-trivial, the question is whether behavior is a *better* predictor of selectivity than other representational schemes -- otherwise, the claim of "behavioral relevance" becomes somewhat weak: Even activation on the retina is "behaviorally relevant" as without retinal input, vision admittedly becomes somewhat difficult. What is missing to give the claim of "behavioral relevance" more heft is a demonstration that the "behaviorally-relevant" axes provide a better description than shape-based ones (that correspond to the current understanding of visual cortex organization), such as "orientation" in V1. Yet, wavelet-based (i.e., orientation-based) V1 encoding models have been shown to provide fits an order of magnitude or more better (see, e.g., Vu et al., *Ann Appl Stat*, 2011) than the "behavioral-centric" representation of Contier et al. Relatedly, the case for labeling some of Contier et al.'s axes as "behaviorally relevant" is at least debatable. For instance, line 201 refers to "grid/grating-related" and "repetitive/spiky" axes, yet those aren't really behaviors but rather shape descriptors. Even "tool-related/handheld/elongated" (l. 199) isn't, unless "tool-related" (which could be called behavior-related) can be shown to be superior to "elongated" (shape-related).

Additionally weakening the support for the claimed "behavior-centric view on visual processing in the human brain", the lack of homogeneity of correlations in areas shown by many prior studies to exhibit clear object selectivity is somewhat surprising: PPA, FFA, OFA, EBA etc. all generally have rather inhomogeneous prediction accuracy, and show non-negligible correlations only in subsections of the ROI, raising the question of why the authors' analyses produce such patchy fits that do not map well

onto known category-selective areas (whose selectivity is thought of as rather homogeneous, see, e.g., the recordings from face patches in monkeys by Tsao et al.). Of further concern in connection with the "map" claim, there seems to be significant variability of correlation patterns with respect to known object-selective areas across the three different subjects (Fig. 2) -- the manuscript's claim that "our findings were highly replicable across the three participants" notwithstanding. What analyses is this latter statement based on?

In summary, the evidence for the claimed "behavior-centric view on visual processing in the human brain" does not seem to be as strong as one would like.

Reviewer #3:

Remarks to the Author:

This paper tackles an important question in visual neuroscience: how is information about behaviorally-relevant visual dimensions represented in the cortex? The authors analyze a large-scale dataset of human similarity judgements to extract behaviorally-relevant dimensions and subsequently map these dimensions onto brain responses (all part of the THINGS data collection) via linearized encoding models. The authors find that these dimensions are predictive of brain responses throughout the visual cortex, going beyond the high-level cortex into earliest stages of visual processing. The authors subsequently interpret the weights of the encoding model to study how tuning for different behavioral dimensions is distributed throughout the cortex. They show that category-selective clusters exhibit sparse tuning profiles with respect to these dimensions. The authors further demonstrate that an encoding model derived from behaviorally-relevant dimensions better predicts responses throughout the visual cortex than an encoding model derived from object category labels. Given these and other analyses, the authors conclude that the behaviorally-relevant dimensions are crucial to how visual information is represented throughout the cortex.

The paper is well-written and addresses an important and topical question, given the widespread interest in understanding the principles of cortical organization and extensive debate in the field regarding the nature and functional significance of category-selective responses. Moreover, the general approach taken in this paper - of systematically and extensively studying the link between behaviorally-relevant dimensions and activity patterns throughout the visual cortex - is interesting. However, it is not entirely clear to what extent the paper provides a novel account of cortical organization or neural tuning. Further, I have some concerns about the analyses and subsequent inferences that reduce my enthusiasm for this work, as outlined below.

It would help if the authors could clearly specify which figures support the result that the behaviorally-relevant dimensions inferred in the study have a distributed cortical representation. On page 4, the authors reference fig. 2 as speaking to these results but the figure only shows prediction accuracies of the encoding model derived from behaviorally-relevant dimensions. How does it support a distributed representation view? It would help to clearly define the 'distributed representation' view in this context. Does it refer to information being spread beyond the high-level visual cortex to early stages of processing? If so, this is different from the traditional view of distributed coding. Even so, while Fig. 2 provides hints that prediction accuracies using the behaviorally relevant dimensions as features of the encoding model are more or less uniform throughout the visual cortex, it would help if the results for different brain regions could be summarized quantitatively (for instance, in bar charts).

The authors mention that the results are expected to be biased in favor of the categorical model but it is not obvious why that is the case. It is not necessary that more dimensions would necessarily lead to higher prediction accuracies and it depends on various factors like the number of data points used for fitting the linearized encoding model etc. I would suggest the authors tone down this claim.

Beyond Fig. 6, the authors should report the amount of variance explained by the category model and the behavioral-dimension model clearly in the results to reveal the extent of the difference between the two.

Further given that some of the dimensions were clearly related to categories, like the 'head-related' dimension or the 'body-related' dimension, isn't it expected that the dimension-based model, by virtue of encompassing the relevant categories, could outperform the categorical model which might not have

faces or bodies as explicit categories. This is particularly relevant for predicting responses in category-selective clusters like FFA or EBA.

Relatedly, the evidence of sparse tuning with respect to behavioral dimensions in category-selective clusters is also perhaps not that surprising given that there are distinct dimensions corresponding to features like 'head-related' or 'body-related'. However, it is interesting that the authors could use the idea of sparseness to identify novel functionally selective regions in a more data-driven manner.

Author Rebuttal to Initial comments

Reviewer #1:

Generalizability of Results: The study's findings are specific to a single dataset and behavioral methodology, with the generalizability of its findings still unknown. Future research could benefit from validating these results across different populations and using varied datasets to ensure broader applicability.

We thank Reviewer 1 for highlighting the potential value of a validation across populations and extending the results to additional datasets. Indeed, we had briefly discussed these issues in our limitations section. While we chose the THINGS-fMRI dataset specifically because its breadth of stimuli makes it specifically suited to link brain and behavioral object responses, to address the reviewer's concerns, we decided to test the degree to which our results would generalize to another dataset and additional participants. In the revised manuscript, we now include additional results where we used behavioral dimensions to predict responses in the large-scale fMRI dataset BOLD5000 which is based on different sets of object and scene images (ImageNet and MS CoCo). Our results replicated successfully, highlighting the generalizability of these findings to the types of natural images used in this other dataset and to additional participants. In the revised manuscript, we now include results showing that the representation of the model dimensions is similarly distributed across voxels in BOLD5000 and that the tuning maps of individual object dimensions are highly similar. (Suppl. Fig. 2).

Line 160: *"We also tested the replicability of these results on an independent fMRI dataset⁶⁷, revealing a similarly extensive representation of the object dimensions (Suppl. Fig. 2)"*

Line 223: *"Importantly, the functional topographies of most object dimensions were also found to be highly consistent across the three subjects in this dataset (Suppl. Fig. 4) and largely similar to participants of an independent, external dataset (Suppl. Fig. 2), suggesting that these topographies may reflect general organizing principles rather than idiosyncratic effects (Suppl. Fig. 4, Extended Data Fig. 1-6)."*

Line 449: “Additionally, applying our approach to an external fMRI dataset (Suppl. Methods. 1) revealed similarly distributed responses, with highly similar dimension tuning maps, suggesting that our findings generalize to independent participants (Suppl. Fig. 2)”

CLIP Pretraining Biases: The use of the CLIP-VIT model to encode images into the 66-dimensional behavioral embedding might introduce additional biases in the embedding due to the language/image pretraining of the model that are not explored in this work. The authors also don’t perform any validation against training data leakage, as it is possible some of the images in the THINGS dataset were part of the training dataset for CLIP, and might have polluted some of the results.

We thank Reviewer 1 for raising this concern. We agree that a CLIP pretraining bias could potentially affect the prediction performance of the behavioral similarity embedding. However, if any, this could only reduce the performance of the encoding model, since the model is evaluated on brain data, not images. A post-hoc analysis in our initial submission (Suppl. Fig. 1), however, suggests that the encoding model performance is consistently improved by the CLIP-based prediction of image-wise dimension values across relevant cortical areas, which confirms that these results generalize to the prediction of patterns of brain activity. At the same time, despite the excellent overall predictive performance, we believe this prediction approach is not going to be perfect compared to empirically measured dimension values for each image. We have added this limitation to the revised Discussion.

Line 459: “Moreover, our use of a pre-trained computational model to obtain predicted dimension values might have underestimated the performance of the object embedding in predicting brain responses or may have selectively improved the fit of some dimensions more than that of others. Future studies could test if using empirically measured dimension values for each image would lead to refined dimension maps.”

[T]his paper represents a significant contribution to our understanding of the human visual system. Its strengths far outweigh its weaknesses, offering a fresh perspective on object vision. The innovative approach, comprehensive data analysis, and insightful findings make it a valuable addition to the field. The weaknesses identified provide opportunities for further research and refinement, which could enhance the impact and applicability of this important work. Overall, I fully recommend this paper be accepted with no hesitations.

We would like to thank Reviewer 1 for their positive overall evaluation of our work and their insightful comments which we believe further strengthened our manuscript.

Reviewer #2:

[T]he present manuscript builds on the very interesting prior behavioral work by the two senior authors, who had analyzed a vast behavioral data set to isolate a set of dimensions that describe human similarity judgments for a large object database (the THINGS database). The objective of the current manuscript was to analyze fMRI responses to a (large) subset of these objects and to see whether the behavioral dimensions were useful as a representational space also for the objects' neural representations across visual cortex. If successful, this would present a major advance in our understanding of the organization of object representations in cortex.

We thank Reviewer 2 for their enthusiasm about the research and emphasizing the impact of our potential findings.

Yet, this question of model fit is a major concern with the manuscript: The main figures in the manuscript focus on the "noise ceiling corrected R^2 ". This raises the critical question of what the noise ceiling is: what is the denominator used to normalize the correlation/explained variance values? Surprisingly, this information is missing from the manuscript. Suppl. Figure 2 appears to show a maximum correlation of 0.2 of predicted and held-out data (without noise ceiling normalization), corresponding to a maximum explained variance of only 4%, with most values appearing considerably lower. The authors need to provide this information on the noise ceiling to make it possible to assess model fit, as they did in their 2023 eLife paper.

We thank Reviewer 2 for their thoughtful comments and suggestions. In the revised manuscript, we now explain how the noise ceiling was computed, which was used to normalize explained variance. The denominator (noise ceiling) was computed by dividing the between-condition variance, estimated between different images, by the within-condition variance, estimated within repeats of the same image (Allen et al., 2022; Hebart et al., 2023). This approach is largely equivalent to computing, for each voxel, the Spearman-Brown-corrected split-half reliability across betas (Huth et al., 2012) but offers an analytical solution.

Line 610: *“We computed noise ceiling corrected R^2 values by dividing the original R^2 of the model by the noise ceiling estimates, for each voxel separately. These single-trial noise ceilings (Suppl. Fig. 7) were provided with the fMRI dataset and were computed based on estimates of the signal and noise variance obtained based on the variability of responses to repeatedly presented images⁵⁷.”*

In addition, in response to the reviewer’s comment, we have added cortical flat maps showing the noise ceiling based on single trial fMRI responses, which yielded noise ceilings of 10% in early and some higher-level visual regions (Suppl. Fig. 7).

Supplementary Fig. 7. **Noise ceiling of single trial responses provided by the THINGS-fMRI dataset.** Colors indicate the noise ceiling expressed as the amount of explainable variance in trial-wise fMRI response estimates which was used to normalize the prediction performance of the encoding model.

Suppl. Figure 2 suggests a rather low predictive power of their scheme, both absolutely and also relatively to Huth et al., which found that semantic axes obtained from the responses of all but one subject were able to account (in the held-out subject) for about 10 times the variance that Contier et al.'s model can explain. Granted, Huth et al.'s representation was data-driven, but the fact remains that the explained variance by Contier et al. appears quite minuscule.

We thank the reviewer for this comment and appreciate the opportunity to compare our results to those presented in the seminal study by Huth et al. (2012). We believe that there are key methodological differences that make a direct comparison misleading and that a direct quantitative comparison with the predictivity in other studies is challenging.

First, the reported model predictivity in Huth et al. was, in fact, also normalized by a noise ceiling (see their Supplementary Material), but they did not report the size of the denominator, preventing a direct comparison of signal-to-noise ratio between both datasets. In our reading of the literature, normalization by the noise ceiling appears to be rather common practice when interpreting encoding model predictivity on fMRI data, as the signal-to-noise ratio can vary drastically between brain regions. Indeed, many studies do not even report this correction in the main findings. We aimed at being transparent about this correction and thus also reported the raw, uncorrected values in the supplement in our original submission.

Second, model predictivity is heavily influenced by the number of repeated samples in the test set. Huth et al. used 10 repetitions to average over in their test set, thus achieving a rather clean dataset for model assessment, which boosts apparent predictive performance. In contrast, our predictions were evaluated on single-trial responses. To allow for a better comparison of noise-ceiling corrected predictions, we re-evaluated our model predictions on the average response across a small, separate subset of images for which 12 repetitions were available (see Response Figure 1 below), which had been collected across scanning sessions. With this adapted measure, our results now explain more than 80% of the explainable variance in some regions. To further evaluate the robustness of our model, we visualized the relationship between model fit and noise ceiling for both single trial and average response scenarios (see Response Figure 2 below). We find that in both cases the model fits better in voxels with higher noise ceilings, indicating that our results are not systematically affected by data reliability.

Response Figure 1. **Prediction accuracy of the object embedding model evaluated on the average test set (12 repetitions).** Results from example subject 01 [noise-normalized R^2].

Response Figure 2. **Relationship between model fit and noise ceiling.** 1. Left: Model evaluated on single-trial responses. Right: Model evaluated on average test set response. Data from example subject 01.

The third key difference is the nature of the object dimensions in Huth et al., which puts the amount of variance they explain in perspective. We used a-priori defined object dimensions to directly predict brain responses, which amounts to variance explained in (raw) brain responses. In contrast, the semantic dimensions in Huth et al. were obtained by identifying

principal components in voxel-wise regression weights (i.e. betas) of an encoding model with thousands of regressors (see below), which amounts to variance explained in the beta estimates. Beta estimates represent a much cleaner transformation of the data which, while tailored to the data-driven approach in Huth et al., cannot be directly compared in terms of explained variance to raw brain data.

Quantitative comparability of our results aside, the encoding model used in Huth et al.'s approach had a fundamentally different purpose to the one we present here. Specifically, we did not aim to explain as much variance in neural responses as possible, but to evaluate how much variance is explained by a purely behavior-driven model. This is also reflected in the vastly different number of regressors, which was much more extensive in Huth et al. with 3,844 regressors (1,705 object/action categories + 2,139 motion energy filters) compared to our 66 object dimensions. We believe that the amount of variance we can explain (up to 50 %) with a much more conservative model of visual responses is, in fact, striking.

Together, these points highlight the overall high predictive performance of our model when taking the noise in the data into account and that our model predictivity, at the level of noise-ceiling corrected R^2 , is comparable to previous findings. In addition, they underscore challenges in comparing results across different datasets (e.g. Huth et al.) or even across modalities (e.g. macaque single unit responses), specifically when trying to generalize results obtained when noise ceilings are low. We agree that this is a highly relevant topic and have added this challenge to the revised Discussion section.

Line 463: *"Finally, we reported results based on noise-ceiling corrected R^2 values. While noise-ceiling normalization is common practice when interpreting encoding model results in order to make them more comparable, the degree to which the results would generalize if noise ceilings were much higher could likely only be addressed with much larger, yet similarly broad datasets."*

The authors stress that their analyses reveal "distributed tuning maps" (l. 374), with activity patterns "mirroring" behaviorally-relevant information. However, even apart from the low % variance explained (that would suggest use of a more cautious term than "mirroring"), the "maps" just seems like a lot of blobs, with little apparent structure.

We thank the reviewer for this insightful comment. We now realize that our wording may have been ambiguous. We did not mean to say that the selectivities to our object dimensions reflect continuous tuning gradients across the cortex (although for some dimensions this may

indeed be the case). Instead, we aimed at highlighting that results are not limited to predefined functionally-selective clusters but extend beyond these regions, with a complex yet stable pattern of responses across voxels. In addition, our intention was to highlight how functionally-selective clusters can emerge from a sparse multidimensional representation. To reduce this ambiguity, we have made two changes to the manuscript.

First, we have removed the phrase “*in the form of distributed tuning maps*” from this section (line 412) and rephrased “*continuous topographic maps*” to “*topographic tuning*” in the introduction (line 57). Second, to avoid confusion for the reader, in the results section of the revised manuscript we are now more explicit about our use of the term tuning maps.

Line 162: “*Please note that, in the following, we use the terms “widespread” and “distributed” interchangeably and do not refer to a distributed representational coding scheme or the presence of continuous gradients but rather to responses that are not locally confined.*”

A core claim of the manuscript is that the study results suggest a "behavior-centric view on visual processing in the human brain." Yet, to make this statement non-trivial, the question is whether behavior is a *better* predictor of selectivity than other representational schemes -- otherwise, the claim of "behavioral relevance" becomes somewhat weak: Even activation on the retina is "behaviorally relevant" as without retinal input, vision admittedly becomes somewhat difficult. What is missing to give the claim of "behavioral relevance" more heft is a demonstration that the "behaviorally-relevant" axes provide a better description than shape-based ones (that correspond to the current understanding of visual cortex organization), such as "orientation" in V1. Yet, wavelet-based (i.e., orientation-based) V1 encoding models have been shown to provide fits an order of magnitude or more better (see, e.g., Vu et al., Ann Appl Stat, 2011) than the "behavioral-centric" representation of Contier et al.

We thank the reviewer for this suggestion. We believe that our initial wording in the manuscript, specifically in the abstract, which was our only use of the term “behavior-centric”, was misleading. Our aim was not to replace stimulus-driven selectivity as a useful framework for visual cortex function. The focus of our contribution was (1) to offer an alternative to a category-based view, which has been dominating the reasoning about the functional significance of the ventral visual stream, and (2) to demonstrate that responses reflecting behaviorally-relevant dimensions, including high-level category-related dimensions, transcend higher-level visual cortex and are found even at the earliest processing stages. The latter contribution is also meant as a change in philosophy, where image-specific responses are not treated as a confound that doesn't generalize to other

images of the same category, but instead as an integral part of the object's representation. It would be interesting to see whether this alternative view would extend to early visual cortex and orientation selectively, but this was not our claim, and even if early visual cortex organization was better explained by behavioral relevance, we do not think that the dimensions would be sufficient for capturing all behaviorally-relevant variance. In the revised manuscript, we now remove the term "behavior-centric" (line 27) and better clarify the reasoning of our work. In addition, in response to another comment of the reviewer below, we will discuss the meaning of the term "behaviorally-relevant", which we realize was similarly ambiguous, not only in the introduction and discussion section of our initial submission, but also in the results section (line 117).

While our results revealed the superiority of behaviorally-relevant dimensions over and above high-level category at capturing visual cortex responses, we had not tested the relevance of object shape. To address this issue, we compared a computational model of perceived object shape to the behaviorally-relevant dimensions in predicting fMRI responses, using an additional variance-partitioning analysis. As a first step, we segmented the images shown in the experiment from their natural background. Next, we ran these segmentations through a performant image-computable model that has been shown to be highly predictive of perceived shape similarity (Morgenstern et al., 2021; see our Suppl. Methods 2). This led to 22 latent shape dimensions underlying more than 100 shape descriptors (e.g. fourier descriptors, major axis orientation, shape skeleton) which we ran through an encoding model. The variance partitioning revealed that, while the two models jointly predicted a fair amount of variance in extrastriate cortex, the behaviorally-relevant dimensions explained substantially more unique variance than shape across visual cortex (Suppl. Fig. 6, see above), highlighting that our model not only encompasses category-selectivity but also some degree of image-specific feature selectivity.

Line 357: "[...] (for an exploratory comparison with object shape, see Suppl. Fig. 6 and Suppl. Methods 2)."

Relatedly, the case for labeling some of Contier et al.'s axes as "behaviorally relevant" is at least debatable. For instance, line 201 refers to "grid/grating-related" and "repetitive/spiky" axes, yet those aren't really behaviors but rather shape descriptors. Even "tool-related/handheld/elongated" (l. 199) isn't, unless "tool-related" (which could be called behavior-related) can be shown to be superior to "elongated" (shape-related).

We thank the reviewer for highlighting this important point. Similar to the previous point, we

realize that our use of the term “behaviorally-relevant” was not explained well enough. With this term, we did not mean to exclusively refer to behaviors related to the affordance of objects or how we interact with them (even though such behaviors may be included, see below). The triplet odd-one-out task underlying the generation of these dimensions serves as a minimal model of general categorization behavior for any type of categorization. Importantly, however, this is much more general than the high-level category labels captured by human language. It can also include categorization behavior for visual features, including shape descriptors. However, these features are not arbitrary, otherwise there would likely be thousands of dimensions in our model. Instead, the model highlights those dimensions that dominate our choice and categorization behavior about objects. Therefore, the term behaviorally-relevant is used to capture the fact that these dimensions are derived from the types of behavioral judgements that underlie our ability to make sense of our visual world, to generalize, structure our environments, and to communicate our knowledge. We now further clarify this perspective in the revised manuscript.

Line 117: *“Thus, the object dimensions are behaviorally-relevant, in that they support the key factors underlying arbitrary categorization behavior and as such underlie our ability to make sense of our visual world, to generalize, structure our environment, and to communicate our knowledge.”*

Additionally weakening the support for the claimed "behavior-centric view on visual processing in the human brain", the lack of homogeneity of correlations in areas shown by many prior studies to exhibit clear object selectivity is somewhat surprising: PPA, FFA, OFA, EBA etc. all generally have rather inhomogeneous prediction accuracy, and show non-negligible correlations only in subsections of the ROI, raising the question of why the authors' analyses produce such patchy fits that do not map well onto known category-selective areas (whose selectivity is thought of as rather homogeneous, see, e.g., the recordings from face patches in monkeys by Tsao et al.).

We thank the reviewer for bringing up the interesting topic of regional homogeneity and for highlighting that the fits appear to be surprising. In our revised manuscript, we now discuss potential reasons for these seemingly inhomogenous fits. We would like to distinguish between prediction accuracy, reflecting the prediction accuracy of the multidimensional model, and the beta maps of individual dimensions. While the prediction accuracy was, in fact, rather homogenous across many of the classical category-selective regions (Fig. 2), it was indeed less homogenous for scene-selective regions PPA and MPA, as well as for EBA. The definition of the

regions was based on a separate functional localizer as part of the THINGS-fMRI dataset (Hebart et al., 2023), contrasting activation of, e.g., “faces > objects”, on spatially smoothed data, with a specific statistical cutoff, and using the intersection with a group atlas to refine the voxel selection (Julian et al., 2012). All these steps, specifically the spatial smoothing, can affect the spatial extent of the identified ROIs. Since there is no consensus on these and numerous other processing choices impacting the exact location of functionally-defined ROIs, we believe this degree of variation is to be expected. In addition, please note that unsmoothed results on cortical flat maps commonly lead to uneven visualization since a mapping from volume to surface data is always an imperfect estimate, which may also appear to lead to patchier fits (e.g. Deniz et al., 2019; Huth et al., 2012; Popham et al., 2021). One way to demonstrate that these effects are not specific to the model of behaviorally-relevant dimensions fitted here is to inspect the noise ceiling maps (see above, see also Suppl. Fig. 7). Indeed, the local regions of low fits are mirrored in parts of the cortical surface with low noise ceilings, highlighting that the seeming inhomogeneity is not a result of our fitting procedure, but may rather be explained by the factors discussed above. The reviewer may not be referring to the overall prediction accuracy but alternatively to the fits of our individual object dimensions that are somewhat related to high-level categories (e.g. outdoors vs. place selective). While the degree of the fits is also mirrored in the noise ceilings, please note that we did not claim that these dimensions directly reflect individual high-level categories but that individual category selectivity emerges as a combination of them (Fig. 4).

Beyond the spatial extent of the identified ROIs, we believe the issue of regional homogeneity is not always seen as clearcut in the literature, with numerous studies suggesting that functionally selective regions may instead comprise subdivisions with distinct selectivities (Baldassano et al., 2013; Betts & Wilson, 2010; Bracci et al., 2010; Çukur et al., 2016; Korhonen et al., 2017; Pinsk et al., 2009; Weiner & Grill-Spector, 2011; White et al., 2019), which could lead to an inhomogeneous distribution of model fits.

Of further concern in connection with the "map" claim, there seems to be significant variability of correlation patterns with respect to known object-selective areas across the three different subjects (Fig. 2) -- the manuscript's claim that "our findings were highly replicable across the three participants" notwithstanding. What analyses is this latter statement based on?

We thank the reviewer for highlighting this point. We now realize that visualizing the beta patterns on the cortical flat maps was not sufficient for demonstrating the replicability of results across participants. As described in the manuscript, the analyses were conducted in individual subject space. We considered a quantitative comparison where we merge results

on a canonical *anatomical* template; however, this would not do justice to the interindividual *functional* heterogeneity, where it is known that functionally-selective clusters often do not end up in the exact same anatomical location across participants. To address this issue, instead we used the functionally-defined regions of interest as anchor points between subjects, extracted their mean beta pattern, and correlated the resulting beta patterns across regions between all pairs of participants, for each dimension individually. The results quantitatively confirm our claim about the replicability of dimension patterns for most dimensions. Interestingly, they also reveal that some dimensions replicated better than others (e.g. “animal-related”, “food-related”), and some were not replicated at all (e.g. “feminine”). We have now added this analysis to our manuscript and the resulting figure to the supplementary material (Suppl. Fig. 4). Since not all dimensions replicated well, we now mention this explicitly in the revised discussion section and have toned down the statement about replicability, from “highly replicable” to “highly replicable for most dimensions”.

Line 223: *“Importantly, the functional topographies of most object dimensions were also found to be highly consistent across the three subjects in this dataset (Suppl. Fig. 4) and largely similar to participants of an independent, external dataset (Suppl. Fig. 2), suggesting that these topographies may reflect general organizing principles rather than idiosyncratic effects (Suppl. Fig. 4, Extended Data Fig. 1-6).”*

Line 443: *“However, our findings were highly replicable across the three participants for most dimensions, suggesting that these dimensions reflect general organizing principles rather than idiosyncratic effects (Suppl. Fig. 4). Of note, some dimensions did not replicate well (e.g. “feminine (stereotypical)”, “hobby-related”, or “foot- / walking-related”; Suppl. Fig. 4), which indicates that our fitting procedure does not yield replicable brain activity patterns for any arbitrary dimension. Future work may test the degree to which these results generalize to other dimensions identified through behavior.”*

Line 638: *“While our analysis was focused on individual subjects, we also estimated the consistency of the tuning maps of individual dimensions across participants. To this end, we used a number of individually-defined regions of interest as anchor points for quantifying similarities and differences between these maps. First, for each dimension separately, we obtained mean beta patterns across these regions, including early visual retinotopic areas (V1-V3 and hV4) as well as face- (FFA, OFA), body- (EBA), and scene-selective (PPA, OPA, MPA) regions. Face-, body-, and scene-selective regions were analyzed separately for each hemisphere to account for potential lateralized effects, and voxels with a noise ceiling smaller than 2% were excluded from the analysis. Finally, to quantify the replicability across participants, we computed the inter-subject correlation based on these mean beta patterns,*

separately for each dimension (Suppl. Fig. 4).”

Supplementary Fig. 4. **Consistency of average ROI dimension tuning across subjects.** Bar heights show the correlation between two participants' dimension tuning patterns for a given dimension. Tuning patterns were obtained by averaging beta values from the encoding model in 16 ROIs. Bar color indicates the subject pair for which the correlation was computed.

Reviewer #3:

The paper is well-written and addresses an important and topical question, given the widespread interest in understanding the principles of cortical organization and extensive debate in the field regarding the nature and functional significance of category-selective responses. Moreover, the general approach taken in this paper - of systematically and extensively studying the link between behaviorally-relevant dimensions and activity patterns throughout the visual cortex - is interesting. However, it is not entirely clear to what extent the paper provides a novel account of cortical organization or neural tuning. Further, I have some concerns about the analyses and subsequent inferences that reduce my enthusiasm for this work, as outlined below.

We thank Reviewer 3 for pointing out the relevance of these questions of cortical organization for the field and bringing up insightful comments. In response to the reviewer's comments, we have now extended our initial analyses. We believe they have strengthened our manuscript and hope they successfully addressed the reviewer's concerns.

It would help if the authors could clearly specify which figures support the result that the behaviorally-relevant dimensions inferred in the study have a distributed cortical representation. On page 4, the authors reference fig. 2 as speaking to these results but the figure only shows prediction accuracies of the encoding model derived from behaviorally-relevant dimensions. How does it support a distributed representation view? It would help to clearly define the 'distributed representation' view in this context. Does it refer to information being spread beyond the high-level visual cortex to early stages of processing? If so, this is different from the traditional view of distributed coding.

We thank Reviewer 3 for their insightful comments about our manuscript and for bringing up this important distinction. In the revised manuscript, to avoid confusion for the reader, we now define our use of the term "distributed" early on in the results section. Instead of using the term to refer to a distributed coding scheme, which is challenging to infer with local fMRI responses, we used the term "distributed" interchangeably with "widespread", to contrast with the notion of responses that are confined specifically to functionally-selective, localized clusters. In addition, in response to Reviewer 2, we clarify that this term does not refer

exclusively to continuous gradients, but rather to responses that are not confined specifically to functionally-selective, localized clusters. By introducing this clarification early on, we hope this change reduced ambiguity in our manuscript.

line 162: *"Please note that, in the following, we use the terms "widespread" and "distributed" interchangeably and do not refer to a distributed representational coding scheme or the presence of continuous gradients but rather to responses that are not locally confined."*

Even so, while Fig. 2 provides hints that prediction accuracies using the behaviorally relevant dimensions as features of the encoding model are more or less uniform throughout the visual cortex, it would help if the results for different brain regions could be summarized quantitatively (for instance, in bar charts).

We thank the reviewer for this suggestion. We have added bar plots showing the prediction performance across ROIs in Suppl. Fig. 3. This new visualization supports our main findings, showing that the behaviorally-relevant dimensions are predictive of both early- and higher-level visual areas. A comparison of the new ROI summary with the flat maps on the same figure shows how V1-V3 includes many voxels in the periphery that were not visually stimulated. These voxel responses were obviously not well predicted, which may explain smaller effects in these regions.

A

B

Supplementary Fig. 3. fMRI encoding model prediction accuracy and average accuracy in different ROIs. A. Prediction accuracy in statistically significant voxels ($p < 0.01$, FDR-corrected). Each row shows flattened cortical surfaces for each subject. Colors indicate Pearson correlation between predicted and held-out data in a between-session 12-fold cross-validation. **B.** Average prediction accuracy expressed as R^2 in different retinotopic (V1, V2, V4, hV4) and category-selective regions of interest (OFA, FFA, EBA, PPA, MPA, OPA). Error bars indicate 95% confidence intervals of the mean.

The authors mention that the results are expected to be biased in favor of the categorical model but it is not obvious why that is the case. It is not necessary that more dimensions would necessarily lead to higher prediction accuracies and it depends on various factors like the number of data points used for fitting the linearized encoding model etc. I would suggest the authors tone down this claim.

We thank the reviewer for highlighting this issue. Our initial aim was to establish a conservative test of our hypothesis that dimensions can explain more unique variance than category by assigning more regressors to the category model and therefore providing greater flexibility for fitting the data, which was based on the same number of data points in both analyses. However, we agree with the reviewer and believe this issue may distract from the actual argument. We have therefore removed this claim from the revised results section.

Beyond Fig. 6, the authors should report the amount of variance explained by the category model and the behavioral-dimension model clearly in the results to reveal the extent of the difference between the two.

We thank the reviewer for this suggestion. We chose flat map visualizations to allow readers to easily see the cortical distribution of the effects all at once. However, we now realize that this may have made it harder to see the exact quantitative difference between the two models. In the results section of the revised manuscript, we have added summary statistics reporting the median and maximum amount of variance explained by either model. In addition, to directly compare the results at a voxel level, we have added scatter plots (Suppl. Fig. 5) that directly compare the variance uniquely explained by a category- and dimension- based model. We believe this shows the pronounced advantage of the dimensions-based model, separate for high- and low-level regions. We have added a corresponding explanation to the Methods section on the variance partitioning analysis.

Line 370: *“The results (Fig. 6) demonstrate that both object dimensions and categories shared a large degree of variance in explaining brain responses, especially in higher-level ventro-temporal and lateral occipital cortices (median = 19%, maximum = 74% shared explained variance) and to a lesser extent in early visual regions (median = 4%, maximum = 19% shared explained variance). This suggests that both models are well suited for predicting responses in the visual system. However, when inspecting the*

unique variance explained by either model, object dimensions explained a much larger amount of additional variance than object categories (Suppl. Fig. 5). This gain in explained variance was not only evident in higher-level regions (median = 10%, maximum = 35% unique explained variance), where both models performed well, but extended across large parts of visual cortex, including early visual regions (median = 8%, maximum = 35% unique explained variance), suggesting that behaviorally-relevant dimensions captured information not accounted for by categories. Conversely, category membership added little unique explained variance throughout the visual system (median = 1 %, maximum = 11%), reaching larger values in higher-level regions (median = 2%, maximum = 11% unique explained variance).”

Line 762: “We also visualized the relationship between the performance of both models quantitatively. To that end, we selected voxels with a noise ceiling of greater than 5% in early- (V1-V3) and higher-level (face-, body-, and scene-selective) regions of interest and created scatter plots comparing the variance uniquely explained by the category- and dimensions-based model in these voxels (Suppl. Fig. 5). To summarize the extent of explained variance, we computed median and maximum values for the shared and unique explained variances in these voxels.”

Supplementary Fig. 5. **Comparison of variance in neural responses uniquely explained by object category vs. dimensions.** Each sample represents one voxel. The x-axis indicates the amount of variance explained by an encoding model of object category, and the y-axis in turn by a model of behaviorally-relevant dimensions.

Voxels above the dashed identity line were better explained by the dimensions model. Color indicates whether voxels belong to early-visual (V1-V3) or higher-level (face-, body-, and scene-selective) regions of interest.

Further given that some of the dimensions were clearly related to categories, like the ‘head-related’ dimension or the ‘body-related’ dimension, isn’t it expected that the dimension-based model, by virtue of encompassing the relevant categories, could outperform the categorical model which might not have faces or bodies as explicit categories. This is particularly relevant for predicting responses in category-selective clusters like FFA or EBA. Relatedly, the evidence of sparse tuning with respect to behavioral dimensions in category-selective clusters is also perhaps not that surprising given that there are distinct dimensions corresponding to features like ‘head-related’ or ‘body-related’.

We thank the reviewer for this insightful comment. We agree that our object embedding model identified dimensions from human similarity judgments that are related to high-level categories, and indeed, the dimensions were able to accurately predict these and many additional categories (Hebart et al., 2020). Please note, however, that the dimensions reflect continuous, graded responses, that transcend binary categories. For example, rather than coding the presence or absence of an animal, the animal-related dimension may reflect the degree to which an object is referred to as being animate, including inanimate objects such as a hobby horse or a robot. Thus, while our model of behaviorally-relevant dimensions may encompass categorical responses, we asked whether they offer a predictive advantage over a traditional, category-based account of visual cortical responses. Our initial results confirm this view, showing that a dimension-based view can account for much more variance in higher visual cortex than high-level categories.

We also thank the reviewer for bringing up the special role of faces and body parts for cortical selectivity. While they are not traditionally considered high-level categories, we agree that their inclusion is highly relevant for a direct comparison with category-selective clusters. As a result, we have adjusted our model comparison to account for the special role of faces and body parts in cortical responses. Specifically, we manually labeled the fMRI stimuli for the occurrence of faces and body parts and have included these labels as two additional categories in the model comparison. If faces and body parts were able to account for the same variance as the behaviorally-relevant dimensions, there should not be much unique variance left that is explained in these regions. The updated results (Fig. 6), however, were indeed highly similar to the results in the original manuscript without these additional categories: While both models share a large fraction of explained variance, especially in higher-level occipitotemporal cortex, the behaviorally-relevant dimensions explained much more unique variance across visual cortex than object

category. This confirms the conclusions of our initial analyses, both in terms of the overall results pattern and in terms of the magnitude of effects, and extends these results to the categories of faces and body parts.

Line 361: “To account for the known selectivity to faces and body parts, we additionally labeled images in which faces or body parts appeared and included them as two additional categories.”

Line 726: “Lastly, we added two more categories by manually identifying images containing human faces or body parts, respectively. We then compiled an encoding model with 52 binary regressors encoding the high-level categories of all respective objects.”

Fig. 6. Comparison of a continuous dimensional and a categorical model of object responses. Flat maps show the left hemisphere of each subject. Colors indicate the proportion of explained variance (noise ceiling corrected R^2) from variance partitioning. A. Shared variance in single-trial fMRI responses explained by both models. B. Variance explained uniquely by a multidimensional model. C. Variance explained uniquely by a model of object categories.

References

- Allen, E. J., St-Yves, G., Wu, Y., Breedlove, J. L., Prince, J. S., Dowdle, L. T., Nau, M., Caron, B., Pestilli, F., Charest, I., Hutchinson, J. B., Naselaris, T., & Kay, K. (2022). A massive 7T fMRI dataset to bridge cognitive neuroscience and artificial intelligence. *Nature Neuroscience*, *25*(1), 116–126. <https://doi.org/10.1038/s41593-021-00962-x>
- Baldassano, C., Beck, D. M., & Fei-Fei, L. (2013). Differential connectivity within the Parahippocampal Place Area. *NeuroImage*, *75*, 228–237. <https://doi.org/10.1016/j.neuroimage.2013.02.073>
- Betts, L. R., & Wilson, H. R. (2010). Heterogeneous structure in face-selective human occipito-temporal cortex. *Journal of Cognitive Neuroscience*, *22*(10), 2276–2288. <https://doi.org/10.1162/jocn.2009.21346>
- Bracci, S., Ietswaart, M., Peelen, M. V., & Cavina-Pratesi, C. (2010). Dissociable neural responses to hands and non-hand body parts in human left extrastriate visual cortex. *Journal of Neurophysiology*, *103*(6), 3389–3397. <https://doi.org/10.1152/jn.00215.2010>
- Chang, N., Pyles, J. A., Marcus, A., Gupta, A., Tarr, M. J., & Aminoff, E. M. (2019). BOLD5000, a public fMRI dataset while viewing 5000 visual images. *Scientific Data*, *6*(1), 49. <https://doi.org/10.1038/s41597-019-0052-3>
- Çukur, T., Huth, A. G., Nishimoto, S., & Gallant, J. L. (2016). Functional Subdomains within Scene-Selective Cortex: Parahippocampal Place Area, Retrosplenial Complex, and Occipital Place Area. *The Journal of Neuroscience: The Official Journal of the Society for Neuroscience*, *36*(40), 10257–10273. <https://doi.org/10.1523/JNEUROSCI.4033-14.2016>

Deniz, F., Nunez-Elizalde, A. O., Huth, A. G., & Gallant, J. L. (2019). The Representation of Semantic Information Across Human Cerebral Cortex During Listening Versus Reading Is Invariant to Stimulus Modality. *The Journal of Neuroscience: The Official Journal of the Society for Neuroscience*, 39(39), 7722–7736. <https://doi.org/10.1523/JNEUROSCI.0675-19.2019>

Hebart, M. N., Contier, O., Teichmann, L., Rockter, A. H., Zheng, C. Y., Kidder, A., Corriveau, A., Vaziri-Pashkam, M., & Baker, C. I. (2023). THINGS-data, a multimodal collection of large-scale datasets for investigating object representations in human brain and behavior. *eLife*, 12, e82580. <https://doi.org/10.7554/eLife.82580>

Hebart, M. N., Zheng, C. Y., Pereira, F., & Baker, C. I. (2020). Revealing the multidimensional mental representations of natural objects underlying human similarity judgements. *Nature Human Behaviour*, 4(11), 1173–1185. <https://doi.org/10.1038/s41562-020-00951-3>

Huth, A. G., Nishimoto, S., Vu, A. T., & Gallant, J. L. (2012). A Continuous Semantic Space Describes the Representation of Thousands of Object and Action Categories across the Human Brain. *Neuron*, 76(6), 1210–1224. <https://doi.org/10.1016/j.neuron.2012.10.014>

Julian, J. B., Fedorenko, E., Webster, J., & Kanwisher, N. (2012). An algorithmic method for functionally defining regions of interest in the ventral visual pathway. *NeuroImage*, 60(4), 2357–2364. <https://doi.org/10.1016/j.neuroimage.2012.02.055>

Korhonen, O., Saarimäki, H., Glerean, E., Sams, M., & Saramäki, J. (2017). Consistency of Regions of Interest as nodes of fMRI functional brain networks. *Network Neuroscience (Cambridge, Mass.)*, 1(3), 254–274. https://doi.org/10.1162/NETN_a_00013

Morgenstern, Y., Hartmann, F., Schmidt, F., Tiedemann, H., Prokott, E., Maiello, G., & Fleming, R. W. (2021). An image-computable model of human visual shape similarity. *PLoS Computational Biology*, 17(6), e1008981. <https://doi.org/10.1371/journal.pcbi.1008981>

Pinsk, M. A., Arcaro, M., Weiner, K. S., Kalkus, J. F., Inati, S. J., Gross, C. G., & Kastner,

S. (2009). Neural representations of faces and body parts in macaque and human cortex: a comparative fMRI study. *Journal of Neurophysiology*, 101(5), 2581–2600.

<https://doi.org/10.1152/jn.91198.2008>

Popham, S. F., Huth, A. G., Bilenko, N. Y., Deniz, F., Gao, J. S., Nunez-Elizalde, A. O., & Gallant, J. L. (2021). Visual and linguistic semantic representations are aligned at the border of human visual cortex. *Nature Neuroscience*, 24(11), 1628–1636.

<https://doi.org/10.1038/s41593-021-00921-6>

Weiner, K. S., & Grill-Spector, K. (2011). Not one extrastriate body area: using anatomical landmarks, hMT+, and visual field maps to parcellate limb-selective activations in human lateral occipitotemporal cortex. *NeuroImage*, 56(4), 2183–2199.

<https://doi.org/10.1016/j.neuroimage.2011.03.041>

White, A. L., Palmer, J., Boynton, G. M., & Yeatman, J. D. (2019). Parallel spatial channels converge at a bottleneck in anterior word-selective cortex. *Proceedings of the National Academy of Sciences of the United States of America*, 116(20), 10087–10096.

<https://doi.org/10.1073/pnas.1822137116>

Decision Letter, first revision:

14th June 2024

Dear Dr. Contier,

Thank you for your patience as we've prepared the guidelines for final submission of your Nature Human Behaviour manuscript, "Distributed representations of behaviorally-relevant object dimensions in the human visual system" (NATHUMBEHAV-23113914A). Please carefully follow the step-by-step instructions provided in the attached file, and add a response in each row of the table to indicate the changes that you have made. Please also address the additional marked-up edits we have proposed within the reporting summary. Ensuring that each point is addressed will help to ensure that your revised manuscript can be swiftly handed over to our production team.

We would hope to receive your revised paper, with all of the requested files and forms within two-three weeks. Please get in contact with us if you anticipate delays.

If you have not done so already, please alert us to any related manuscripts from your group that are under consideration or in press at other journals, or are being written up for submission to other journals (see: <https://www.nature.com/nature-research/editorial-policies/plagiarism#policy-29>)

on-duplicate-publication for details).

Nature Human Behaviour offers a Transparent Peer Review option for new original research manuscripts submitted after December 1st, 2019. As part of this initiative, we encourage our authors to support increased transparency into the peer review process by agreeing to have the reviewer comments, author rebuttal letters, and editorial decision letters published as a Supplementary item. When you submit your final files please clearly state in your cover letter whether or not you would like to participate in this initiative. Please note that failure to state your preference will result in delays in accepting your manuscript for publication.

In recognition of the time and expertise our reviewers provide to Nature Human Behaviour's editorial process, we would like to formally acknowledge their contribution to the external peer review of your manuscript entitled "Distributed representations of behaviorally-relevant object dimensions in the human visual system". For those reviewers who give their assent, we will be publishing their names alongside the published article.

Cover suggestions

We welcome submissions of artwork for consideration for our cover. For more information, please see our guide for cover artwork.

ORCID

Non-corresponding authors do not have to link their ORCIDs but are encouraged to do so. Please note that it will not be possible to add/modify ORCIDs at proof. Thus, please let your co-authors know that if they wish to have their ORCID added to the paper they must follow the procedure described in the following link prior to acceptance:

Nature Human Behaviour has now transitioned to a unified Rights Collection system which will allow our Author Services team to quickly and easily collect the rights and permissions required to publish your work. Approximately 10 days after your paper is formally accepted, you will receive an email in providing you with a link to complete the grant of rights. If your paper is eligible for Open Access, our Author Services team will also be in touch regarding any additional information that may be required to arrange payment for your article.

Please note that *Nature Human Behaviour* is a Transformative Journal (TJ). Authors may publish their research with us through the traditional subscription access route or make their paper immediately open access through payment of an article-processing charge (APC). Authors will not be required to make a final decision about access to their article until it has been accepted. Find out more about Transformative Journals

[REDACTED]

Best regards,

[REDACTED]

On behalf of

[REDACTED]

Reviewer #1:

Remarks to the Author:

Overall I believe the authors have been thorough in their responses, and my remaining concerns are relatively minor compared to the scope and contributions of their paper. Some additional comments below.

Regarding the generalizability of results: The authors' successful application of their method to a different fMRI dataset (BOLD5000) addresses my primary concern and shows the robustness of their work.

Regarding the pretraining biases of CLIP: There is still an issue here. There is now a published paper showcasing the performance of an encoding model based on CLIP (Wang et al., 2023). The model of the current study is identical to the one in Wang et al., *up to a linear transformation*. That linear transformation —from CLIP to the behavioral similarity embedding space of Hebart et al.—is the main contribution the current paper offers over Wang et al. This linear layer is potentially really important! But, it's hard to know.

The easiest way to address this would be to compare the encoding model of the current paper to one where the regression weights from CLIP to embedding space are replaced with a random linear transformation. This would show if the linear layer mapping from CLIP to behavioral-similarity-embeddings is doing important work that is not done by the CLIP features alone. Without this, there is currently no evidence in the paper to support a claim that the current model is doing or explaining anything that the previously published CLIP model of Wang et al does not.

That said, although the Wang et al. paper does undermine the novelty of the current study, I do not believe my criticism undermines the paper's core claim regarding the improvement of a behavior-derived encoding model over one trained using category labels. CLIP is a model designed to produce embeddings derived from human descriptions of images, which are inherently a kind of behavioral readout. One possible interpretation of these results is that the CLIP model itself is trained to create "behaviorally-aligned" highly-dimensional embeddings from images, which happen to be very good at explaining variance in visual cortex, and that once these embeddings are mapped to the specific behavioral dimensions identified by the authors, a significant part of the useful information contained in those original embeddings is preserved, and thus can still predict a lot of neural variance. Thus, at the very least, the current paper shows that CLIP features can be reduced to 66 dimensions and still explain more variance than an object category model.

Wang, A.Y., et al. Better models of human high-level visual cortex emerge from natural language supervision with a large and diverse dataset. *Nat Mach Intell* 5, 1415–1426 (2023).
<https://doi.org/10.1038/s42256-023-00753-y>

Reviewer #2:

Remarks to the Author:

I greatly appreciated the authors' response to the points raised in my review, and I compliment them on the very informative additional analyses they performed (I'd like to highlight the "consistency of average ROI dimension tuning across subjects" analysis, which is sure to provide a lot of food for thought, as well as their nice discussion of noise ceiling and the effect of stimulus repetition, which I found quite enlightening). In addition, the numerous requested changes in the language have increased the precision of their claims, making the novelty and impact of their findings more readily apparent. Specifically, the removal of "behavior-centric" is appreciated, as the authors' representational scheme is not one based on behaviors, but on human similarity judgments, as discussed. However for the same reason – and this is my only remaining criticism – I would strongly urge the authors to also change the term "behaviorally-relevant", as it is similarly misleading or at least easily misunderstood. I appreciate the authors' point that they want to use "behaviorally-relevant" as a counterpoint to "categorical" representations, and the new clarification of the intended meaning of "behaviorally-relevant" in lines 117-120 is helpful indeed. However, the clarification there that "behaviorally-relevant" refers to factors "underlying arbitrary categorization behaviors and as such underlie our ability to make sense of our visual world, to generalize, structure our environment, and to communicate our knowledge" makes it such a broad term that any kind of image description (even merely shape-based ones) would qualify. The authors' scheme is clearly more than shape-based (as it includes not only shape-based dimensions such as "spiky" or "grating-related", but also dimensions such as "disgusting" and "outdoors" that clearly go beyond mere shape), but "behaviorally-relevant" (as even used in the title) raises the wrong expectations and gets the paper off on the wrong foot. Given the learned readership of Nature Human Behaviour, "human similarity judgment-derived dimensions", though admittedly more of a mouthful, would still be parsable and, crucially, more precise and impactful. After all, the cool thing about the present manuscript is that it shows that the authors' previously derived model of how the mind represents the world can serve as a framework to make progress in understanding cortical object representations.

In any case, that quibble about the squishy, bland & easily misunderstood term "behaviorally-relevant" is my only remaining concern. The authors have done an admirable job in responding to my previous concerns, and I feel the resulting manuscript now will make a very interesting contribution to the question of how objects are represented in cortex -- one that will provide ample food for discussion and move the field forward.

Author Rebuttal, first revision:

Reviewer #1

Overall I believe the authors have been thorough in their responses, and my remaining concerns are relatively minor compared to the scope and contributions of their paper. Some additional comments below.

We thank Reviewer #1 for positively evaluating our response, for highlighting the scope and contribution of our work, and for providing further helpful comments.

Regarding the generalizability of results: The authors' successful application of their method to a different fMRI dataset (BOLD5000) addresses my primary concern and shows the robustness of their work.

We are pleased to hear that the reviewer's primary concern has been successfully addressed. We also want to thank the reviewer again for their initial suggestion to replicate our results in another dataset which we believed has strengthened our findings.

Regarding the pretraining biases of CLIP: There is still an issue here. There is now a published paper showcasing the performance of an encoding model based on CLIP (Wang et al., 2023). The model of the current study is identical to the one in Wang et al., *up to a linear transformation*. That linear transformation –from CLIP to the behavioral similarity embedding space of Hebart et al.-is the main contribution the current paper offers over Wang et al. This linear layer is potentially really important! But, it's hard to know.

The easiest way to address this would be to compare the encoding model of the current paper to one where the regression weights from CLIP to embedding space are replaced with a random linear transformation. This would show if the linear layer mapping from CLIP to behavioral-similarity-embeddings is doing important work that is not done by the CLIP features alone. Without this, there is currently no evidence in the paper to support a claim that the current model is doing or explaining anything that the previously published CLIP model of Wang et al does not.

That said, although the Wang et al. paper does undermine the novelty of the current study, I do not believe my criticism undermines the paper's core claim regarding the improvement of a behavior-derived encoding model over one trained using category labels. CLIP is a model designed to produce embeddings derived from human descriptions of images, which are inherently a kind of behavioral readout. One possible interpretation of these results is that the CLIP model itself is trained to create "behaviorally-aligned" highly-dimensional embeddings from images, which happen to be very good at explaining variance in visual cortex, and that once these embeddings are mapped to the specific behavioral dimensions identified by the authors, a significant part of the useful information contained in those original embeddings is preserved, and thus can still predict a lot of neural variance. Thus, at the very least, the current paper shows that CLIP features can be reduced to 66 dimensions and still explain more variance than an object category model.

We thank the reviewer for highlighting the potential relationship between our findings and those of Wang et al. (2023). We agree that, by applying our dimension-prediction approach, the behavior-derived dimensions can be seen as a 66-dimensional linear transformation of CLIP features. In that sense, the approach shows that CLIP carries information sufficient for characterizing much of the variance in the behavior-derived dimensions.

However, we would like to highlight (1) that our use of CLIP had a very different aim than that of Wang et al., and (2) that it is unlikely that our results reflect a more or less arbitrary mapping of CLIP to brain data rather than the behavior-derived dimensions. Regarding the aim, the purpose of our work was not to use CLIP *as a model* to explain as much variance in visual responses as possible but whether we could use CLIP *as a tool* to interpolate dimension values for images that had not been included in the original similarity task. Thus, we intentionally reduced the expressive power of CLIP to 66 one-dimensional projections, with the aim of potentially improving predictive performance in the brain data. We thus believe that the fact that this 66 dimensional space could potentially be construed as a subspace of CLIP is perhaps of less theoretical interest and should only be seen as a methodological necessity.

While we very much liked the suggestion of the reviewer for testing arbitrary mappings of CLIP to 66 dimensions, we believe our results already demonstrate that it is very unlikely that our findings are merely an artifact of using CLIP for the prediction. As highlighted in Supplementary Figure 1, even without the use of CLIP, our results show strong predictive performance across

voxels. This result would not be possible if our results were an artifact of using CLIP. In addition, there is a very strong correlation between predictive performance *before* the use of CLIP with performance *after* the use of CLIP. This indicates that the fit likely only led to a quantitative difference, not a qualitative difference. Thus, we believe the use of CLIP does not undermine the novelty of our results. We now clarify our rationale and cite the work by Wang et al. in the revised results section:

Line 136: "This model has previously been shown to provide a good correspondence to behavioral (Muttenthaler et al., 2023; Muttenthaler & Hebart, 2021) and brain data (Conwell et al., 2023; Wang et al., 2023), indicating its potential for providing accurate image-wise estimates of behavior-derived object dimensions."

We would further like to emphasize that the key findings that could possibly be affected are (1) the spatial distribution of the fit and (2) the model comparison between the dimension and category model. To fully demonstrate that our findings are not undermined by the use of CLIP, we re-ran these analyses without the use of CLIP but with the dimension values for each object category. As can be seen below, and as expected, the results are weaker - specifically in early visual cortex, since we are no longer fitting an image-specific model, and thus, the model can only rely on visual effects shared between images of a category (Response Fig. 1). Note, however, that the effects in early visual cortex are still present, which is also confirmed by the correlation analysis of predictive performance before and after the use of CLIP (see above). Similarly, the comparisons with the category model yield largely similar results without the use of CLIP yet with slightly weaker effects (Response Fig. 2).

Response Fig. 1. **Prediction accuracy of object dimensions obtained without using CLIP.** As in the main analysis, core object dimensions were obtained from behavioral odd-one-out judgements. CLIP was not used to obtain predicted dimension values for each image. The fitting procedure was analogous to our main analysis. Colors indicate the proportion of explained variance (noise ceiling corrected R2) of held-out data in a 12-fold between-session cross-validation

Response Fig. 2. **Model comparison of object categories and behavior-derived dimensions without using CLIP.** Colors indicate the proportion of explained variance shared by both models (A), uniquely explained by object dimensions (b), and uniquely explained by a model of object categories (noise ceiling corrected R^2). Model construction, fitting procedure, and variance partitioning were analogous to the analysis in the original manuscript, except the dimension model is only based on empirically measured odd-one-out responses and CLIP was not used to obtain image-wise dimension predictions.

Reviewer #2 (Remarks to the Author):

[...] However for the same reason – and this is my only remaining criticism – I would strongly urge the authors to also change the term "behaviorally-relevant", as it is similarly misleading

or at least easily misunderstood. I appreciate the authors' point that they want to use "behaviorally-relevant" as a counterpoint to "categorical" representations, and the new clarification of the intended meaning of "behaviorally-relevant" in lines 117-120 is helpful indeed. However, the clarification there that "behaviorally-relevant" refers to factors "underlying arbitrary categorization behaviors and as such underlie our ability to make sense of our visual world, to generalize, structure our environment, and to communicate our knowledge" makes it such a broad term that any kind of image description (even merely shape-based ones) would qualify. The authors' scheme is clearly more than shape-based (as it includes not only shape-based dimensions such as "spiky" or "grating-related", but also dimensions such as "disgusting" and "outdoors" that clearly go beyond mere shape), but "behaviorally-relevant" (as even used in the title) raises the wrong expectations and gets the paper off on the wrong foot. Given the learned readership of *Nature Human Behaviour*, "human similarity judgment-derived dimensions" , though admittedly more of a mouthful, would still be parsable and, crucially, more precise and impactful. [...]

We thank the reviewer for their comments. In response, we have adjusted the language in our manuscript to provide more nuance. We now distinguish "behaviorally-relevant" as the phenomenon we want to explain from "behavior-derived" as an attribute of the specific model we used. As a result, we now use "behaviorally-relevant" much more sparsely throughout the manuscript and replaced it in the title with "behavior-derived". When we use "behaviorally-relevant", then only after having clearly defined what we mean by it.

As a consequence, we did not remove the term "behaviorally-relevant" completely. While it is a broad term, we still find it useful. As highlighted in our manuscript, we believe that the model we use is well suited to identify many such behaviorally-relevant responses because it captures much of the information we use to distinguish between objects, yielding a representation that not only reproduces perceived similarity and categorization, but could in principle support much more diverse behavioral tasks. We believe with the adaptation, the previous ambiguity should now be resolved.

Final Decision Letter:

Dear Mr Contier,

We are pleased to inform you that your Article "Distributed representations of behavior-derived object dimensions in the human visual system", has now been accepted for publication in *Nature Human Behaviour*.

Please note that *Nature Human Behaviour* is a Transformative Journal (TJ). Authors may publish their research with us through the traditional subscription access route or make their paper immediately open access through payment of an article-processing charge (APC). Authors will not be required to make a final decision about access to their article until it has been accepted. Find out more about Transformative Journals

Authors may need to take specific actions to achieve compliance with funder and institutional open access mandates. If your research is supported by a funder that requires immediate open access (e.g.

according to Plan S principles) then you should select the gold OA route, and we will direct you to the compliant route where possible. For authors selecting the subscription publication route, the journal's standard licensing terms will need to be accepted, including self-archiving policies. Those licensing terms will supersede any other terms that the author or any third party may assert apply to any version of the manuscript.

With best regards,
[REDACTED]